# Symmetry-Aware Bayesian Optimization via Max Kernels

**Anthony Bardou**
School of Computer and Communication Sciences
EPFL
Lausanne, Switzerland
`anthony.bardou@epfl.ch`

**Antoine Gonon**
Institute of Mathematics
EPFL
Lausanne, Switzerland
`antoine.gonon@epfl.ch`

**Aryan Ahadinia & Patrick Thiran**
School of Computer and Communication Sciences
EPFL
Lausanne, Switzerland
`{aryan.ahadinia,patrick.thiran}@epfl.ch`

## Abstract

Bayesian Optimization (BO) is a powerful framework for optimizing noisy, expensive-to-evaluate black-box functions. When the objective exhibits invariances under a group action, exploiting these symmetries can substantially improve BO efficiency. While using maximum similarity across group orbits has long been considered in other domains, the fact that the max kernel is not positive semidefinite (PSD) has prevented its use in BO. In this work, we revisit this idea by considering a PSD projection of the max kernel. Compared to existing invariant (and non-invariant) kernels, we show it achieves significantly lower regret on both synthetic and real-world BO benchmarks, without increasing computational complexity. **Code.** github.com/abardou/max-kernel.

## 1 Introduction

Many real-world problems can be framed as the optimization of a noisy, expensive-to-evaluate black-box function $f^\star : \mathcal{S} \subset \mathbb{R}^d \to \mathbb{R}$. Bayesian Optimization (BO) provides a principled and sample-efficient framework for tackling this problem, with asymptotic guarantees of global optimality complementing its empirical success. As a result, BO has been widely adopted across diverse domains such as robotics (Lizotte et al., 2007), computational biology (Gonzalez et al., 2015) and computer networks (Bardou et al., 2025).

For a black-box function $f^\star$ belonging to the Reproducing Kernel Hilbert Space (RKHS) $\mathcal{H}_k$ associated with a kernel $k : \mathcal{S} \times \mathcal{S} \to \mathbb{R}$, BO proceeds by placing a Gaussian Process (GP) prior $f \sim \mathcal{GP}(0, k)$ over functions in $\mathcal{H}_k$. The kernel $k$ determines the covariance structure of the GP and thus encodes prior assumptions about $f^\star$. Incorporating suitable prior knowledge can substantially improve convergence and sample efficiency. In many applications, the objective is known to be invariant under the action of a group $\mathcal{G}$, that is,

$$f^\star(\boldsymbol{x}) = f^\star(g\boldsymbol{x}) \quad \text{for all } g \in \mathcal{G}.$$

For instance, in molecular property prediction, $f^\star$ may be invariant to rotations of the underlying molecular structure (Glielmo et al., 2017). In such cases, designing kernels that explicitly incorporate $\mathcal{G}$-invariance becomes essential. Ginsbourger et al. (2012) showed that for a centered GP to be $\mathcal{G}$-invariant, its covariance function must also be invariant under $\mathcal{G}$. Motivated by this, we revisit a simple idea—*keep the best alignment over each orbit*—and apply it to BO.

Given a base kernel $k_{\mathrm{b}}$ and a symmetry group $\mathcal{G}$, define

$$k_{\max}(\boldsymbol{x}, \boldsymbol{x}') = \max_{g, g' \in \mathcal{G}} k_{\mathrm{b}}(g\boldsymbol{x}, g'\boldsymbol{x}'), \tag{1}$$

so that the similarity between $\boldsymbol{x}$ and $\boldsymbol{x}'$ is the best alignment over their orbits.

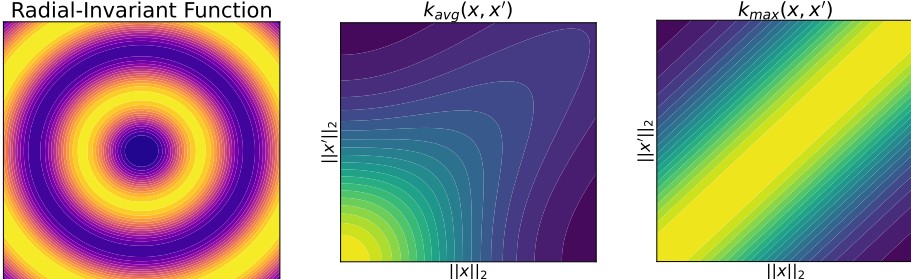

Figure 1: (Left) Colored level sets of a two-dimensional function $f^\star(\boldsymbol{x})$ invariant under planar rotations (see (16)): if $\|\boldsymbol{x}\|_2 = \|\boldsymbol{x}'\|_2$, then $f^\star(\boldsymbol{x}) = f^\star(\boldsymbol{x}')$. (Center/Right) Rotation-invariant kernels derived from an RBF base kernel (lengthscale $1/2$), visualized as a function of $(\|\boldsymbol{x}\|_2, \|\boldsymbol{x}'\|_2)$. $k_{\max}$ (right) captures the correct invariance, while $k_{\mathrm{avg}}$ (center) only approximates it.

The intuition for using the max-alignment is that when the objective is invariant under a group of transformations, two inputs can become very similar *after* applying the right group element, even if they differ a lot in their original positions. For instance, in an image-based problem with rotation invariance, two rotated images of the same object (e.g., cats) should in principle be treated similarly by the optimizer since they correspond to the same objective value. However, most rotations will not align the images well; and if the optimizer compares images with $\ell^2$ distances, only a small number of them can give a good match. In such settings, taking the *maximum* similarity over all group actions is natural: among all transformations, typically only one (or a few) reveal a true alignment. Averaging over all rotations would dilute this information—most transformed pairs look different—whereas the max retains the one transformation that matters. This "best-alignment" principle is the core motivation behind $k_{\max}$ and is expected to provide a clearer signal to the optimizer about which inputs should be treated similarly, compared, e.g., to an averaging approach.

While $k_{\max}$ is symmetric and $\mathcal{G}$-invariant, it is however not guaranteed to be positive semi-definite (PSD), a property required for the standard Gaussian-process machinery underlying BO (see Appendix C.1). To address this, we introduce a PSD version of $k_{\max}$.

**A PSD, invariant surrogate via projection + Nyström.** On a finite design set $\mathcal{D}$, we form the Gram matrix of $k_{\max}$ and project it onto the PSD cone (eigenvalue clipping), obtaining $\boldsymbol{K}_+$. Denoting by $\boldsymbol{K}_+^\dagger$ the Moore-Penrose pseudo-inverse of $\boldsymbol{K}_+$, we then define the $\mathcal{G}$-invariant, PSD kernel

$$k_+^{(\mathcal{D})}(\boldsymbol{x}, \boldsymbol{x}') \;=\; k_{\max}(\boldsymbol{x}, \mathcal{D}) \, \boldsymbol{K}_+^\dagger \, k_{\max}(\mathcal{D}, \boldsymbol{x}'). \tag{2}$$

Equivalently, $k_+^{(\mathcal{D})}(\boldsymbol{x}, \boldsymbol{x}') = \phi(\boldsymbol{x})^\top \phi(\boldsymbol{x}')$ with features $\phi(\boldsymbol{x}) = \boldsymbol{K}_+^{\dagger/2} \, k_{\max}(\mathcal{D}, \boldsymbol{x})$, which makes positive semidefiniteness immediate. By construction, $k_+^{(\mathcal{D})}$ (i) coincides with $k_{\max}$ on $\mathcal{D}$ whenever $k_{\max}$ is already PSD, and (ii) has per-iteration asymptotic cost comparable to orbit-averaged kernels; details in Section 3.2.

**Results.** The max-alignment heuristic does translate into concrete benefits for BO, which we observe throughout the paper. The resulting kernel is geometrically better aligned with the true structure of the problem (Figures 1 and 2). In practice, this makes (i) the acquisition function more faithful as it avoids redundant exploration of points that are already explored up to symmetry, and (ii) uncertainty modeling also more faithful: it gains confidence in unexplored regions that correspond to symmetry-equivalent points. Across synthetic benchmarks with finite and continuous groups, a wireless-network design task and a particle packing problem, we show that $k_+^{(\mathcal{D})}$ consistently attains lower cumulative and simple regret than both the base kernel and the orbit-averaged alternative, with gains increasing with $|\mathcal{G}|$.

**Relation with spectral-based theory.** Mainstream BO theory links fast eigendecay of the kernel to small regret upper bounds (Srinivas et al., 2012; Valko et al., 2013; Scarlett et al., 2017; Whitehouse et al., 2023). Surprisingly, we find the opposite trend in our setting: $k_+^{(\mathcal{D})}$ typically has a *slower* empirical eigendecay than $k_{\mathrm{avg}}$, yet consistently achieves *better (lower)* regret in practice. This directly challenges the usual spectral intuition: our results reveal a clear mismatch between spectral predictions and empirical performance, suggesting that eigendecay alone does not capture the advantages of $k_+^{(\mathcal{D})}$. As we discuss later, geometric considerations (the alignment of the kernel

eigenvectors with the directions that matter for optimization) and approximation hardness of the blackbox $f^\star$ in the RKHS likely play an essential role beyond pure spectral rates.

**Summary of the contributions.** We propose $k_{\max}$ as a *max-alignment* route to $\mathcal{G}$-invariance, turn it into a valid GP kernel for BO via PSD projection and Nyström, and show $k_+^{(\mathcal{D})}$ is $\mathcal{G}$-invariant, equals $k_{\max}$ on $\mathcal{D}$ when $k_{\max}$ is PSD, and matches the asymptotic cost of orbit-averaged kernels (Section 3). We demonstrate consistent BO gains over orbit averaging across BO benchmarks (Section 4), and we analyze why eigendecay alone does not explain these gains (Section 5).

## 2 BACKGROUND

### 2.1 BAYESIAN OPTIMIZATION IN A NUTSHELL

**Problem.** We seek to maximize an expensive-to-evaluate, black-box objective $f^\star : \mathcal{S} \to \mathbb{R}$ under the assumption that $f^\star$ is in the RKHS $\mathcal{H}_k$ of a PSD kernel $k : \mathcal{S} \times \mathcal{S} \to \mathbb{R}$. Each query $\boldsymbol{x} \in \mathcal{S}$ returns a noisy observation $y = f^\star(\boldsymbol{x}) + \varepsilon$, where $\varepsilon \sim \mathcal{N}(0, \sigma_0^2)$. Let $\mathcal{Z}_t = \{(\boldsymbol{x}_i, y_i)\}_{i=1}^t$ denote the dataset after $t$ evaluations, and write $\mathcal{D}_t = (\boldsymbol{x}_1, \ldots, \boldsymbol{x}_t)$ and $\boldsymbol{y}_t = (y_1, \ldots, y_t)^\top$.

**Surrogate model: the GP prior.** BO maintains a probabilistic surrogate $f$ over functions in $\mathcal{H}_k$ to guide sampling of new queries $\boldsymbol{x} \in \mathcal{S}$ with the goal of converging to $\arg\max_{x \in \mathcal{S}} f^\star(\boldsymbol{x})$. A common choice is a zero-mean Gaussian process (GP) (Rasmussen & Williams, 2006),

$$f \sim \mathcal{GP}(0, k),$$

Conditionally on the dataset of queried points $\mathcal{Z}_t$ after $t$ evaluations, the posterior $f \mid \mathcal{Z}_t$ is still a GP with posterior mean and covariance

$$\mu_t(\boldsymbol{x}) = k(\boldsymbol{x}, \mathcal{D}_t) \left( \boldsymbol{K}_t + \sigma_0^2 \boldsymbol{I}_t \right)^{-1} \boldsymbol{y}_t, \tag{3}$$

$$\mathrm{Cov}_t(\boldsymbol{x}, \boldsymbol{x}') = k(\boldsymbol{x}, \boldsymbol{x}') - k(\boldsymbol{x}, \mathcal{D}_t) \left( \boldsymbol{K}_t + \sigma_0^2 \boldsymbol{I}_t \right)^{-1} k(\mathcal{D}_t, \boldsymbol{x}'), \tag{4}$$

where $\boldsymbol{K}_t = k(\mathcal{D}_t, \mathcal{D}_t) \in \mathbb{R}^{t \times t}$, $\boldsymbol{I}_t$ is the $t \times t$ identity, and $k(\boldsymbol{x}, \mathcal{D}_t) = [k(\boldsymbol{x}, \boldsymbol{x}_1), \ldots, k(\boldsymbol{x}, \boldsymbol{x}_t)]$.

The GP posterior plays the role of a refined surrogate for $f^\star$ throughout the optimization process. At iteration $t$, a BO algorithm completes the next steps:

**Step 1.** Form the Gram matrix $\boldsymbol{K}_t = k(\mathcal{D}_t, \mathcal{D}_t)$ using all past queries.

**Step 2.** Compute the inverse of $\boldsymbol{K}_t + \sigma_0^2 \boldsymbol{I}_t$ (with fixed hyperparameter $\sigma_0$) and plug it into (3)-(4) to obtain the posterior mean and covariance functions $(\mu_t, \mathrm{Cov}_t)$.

**Step 3.** Select the next query by maximizing an acquisition function $\alpha_t : \mathcal{S} \to \mathbb{R}$ built from $(\mu_t, \mathrm{Cov}_t)$ (e.g., GP-UCB (Srinivas et al., 2012) or Expected Improvement (Jones et al., 1998)). This is where BO balances *exploration* (learning $f^\star$) and *exploitation* (sampling near current optima). The pair $(\mu_t, \sigma_t^2)$ can be viewed as the algorithm's current best estimate of the unknown function and its uncertainty.

The dataset is then updated with the new query:

$$\boldsymbol{x}_{t+1} \in \arg\max_{\boldsymbol{x} \in \mathcal{S}} \alpha_t(\boldsymbol{x}), \qquad y_{t+1} = f^\star(\boldsymbol{x}_{t+1}) + \varepsilon_{t+1},$$

and the loop repeats until a stopping criterion is met.

**Measuring performance with regret.** We follow the common practice in BO: for experiments where $f^\star$ is known, we measure the regret on the deterministic $f^\star \in \mathcal{H}_k$, and when discussing theoretical regret bounds we refer to the regret on $f \sim \mathcal{GP}(0, k)$ (Garnett, 2023). In both cases, for $h = f$ or $h = f^\star$, the *instantaneous regret* at timestep $t$ is $r_t = \max_{\boldsymbol{x} \in \mathcal{S}} h(\boldsymbol{x}) - h(\boldsymbol{x}_t)$, the *cumulative regret* at horizon $T$ is $R_T = \sum_{t=1}^T r_t$, and the *simple regret* is $s_T = \max_{\boldsymbol{x} \in \mathcal{S}} h(\boldsymbol{x}) - \max_{1 \le t \le T} h(\boldsymbol{x}_t)$. A BO algorithm with a sublinear regret (i.e., $R_T \in o(T)$) is called *no-regret* and offers asymptotic global optimization guarantees on $f^\star$. Most standard cumulative regret upper bounds are established in terms of the eigendecay of the operator spectrum of the kernel $k$ (Srinivas et al., 2012; Valko et al., 2013; Scarlett et al., 2017; Whitehouse et al., 2023).

## 2.2 INVARIANCE IN BAYESIAN OPTIMIZATION

In many applications, the objective function $f^\star$ is invariant under the action of a known symmetry group $\mathcal{G}$ on $\mathcal{S}$, i.e., $f^\star(\boldsymbol{x}) = f^\star(g\boldsymbol{x})$ for all $g \in \mathcal{G}$. When such invariances are ignored, BO algorithms may waste evaluations by treating all points within the same $|\mathcal{G}|$-orbit as distinct. Given a non-invariant base kernel $k_\mathrm{b}$ and an arbitrary symmetry group $\mathcal{G}$, both provided by the user, this section reviews existing strategies for incorporating group invariance into BO and positions our contribution within this literature.

**Data augmentation.** A popular way to enforce symmetry is to expand the dataset $\mathcal{Z}$ itself, as it is often done in computer vision (Krizhevsky et al., 2012). For each acquired observation $(\boldsymbol{x}_t, y_t)$, one augments $\mathcal{Z}$ with all transformed copies $\{(g\boldsymbol{x}_t, y_t)\}_{g \in \mathcal{G}}$, while leaving the base kernel $k_\mathrm{b}$ unchanged. However, since BO scales as $\mathcal{O}(|\mathcal{Z}|^3)$, this approach quickly becomes computationally prohibitive and is inapplicable to continuous symmetry groups. For completeness, we include in Appendix F a numerical comparison of our approach with data augmentation, showing that data augmentation scales poorly with the size of the group, and does not meet the performance of the average or max kernel even when using all symmetry augmentations.

**Search space restriction.** Another approach is to restrict the search domain to a fundamental region $\mathcal{S}_\mathcal{G} \subseteq \mathcal{S}$ whose $\mathcal{G}$-orbit covers $\mathcal{S}$: $\bigcup_{g \in \mathcal{G}} g\mathcal{S}_\mathcal{G} = \mathcal{S}$ (e.g., Baird et al. (2023b)). For example, if $\mathcal{S} = [-1, 1]^2$ and $\mathcal{G}$ is the group of $\pi/2$-rotations, one may work on $\mathcal{S}_\mathcal{G} = [0, 1]^2$ while keeping the kernel unchanged. This viewpoint corresponds to working directly with the quotient $\mathcal{S}/\mathcal{G}$ embedded in $\mathcal{S}$.

This line of work is complementary to ours. In BO, one must choose both a search domain and a kernel: fundamental domains address the former, while our construction helps with the latter. Even if we decide to run BO on $\mathcal{S}_\mathcal{G}$, one still needs a good invariant kernel on $\mathcal{S}_\mathcal{G}$, and our invariant kernels can be used in that setting as well. We refer to Appendix G for a short example illustrating the practical difficulties of explicitly optimizing over a fundamental domain, and how the design of the kernel is complementary to that decision.

**Invariant kernels.** A principled way to incorporate prior $\mathcal{G}$-invariance of $f^\star$ is to consider a $\mathcal{G}$-invariant GP prior $f$, i.e., a GP whose sample paths $\boldsymbol{x} \in \mathcal{S} \mapsto f(\boldsymbol{x}, \omega)$ obtained by fixing one outcome $\omega$ in the probability space are themselves invariant under $\mathcal{G}$. Ginsbourger et al. (2012) established that such GPs necessarily admit a $\mathcal{G}$-invariant covariance function[1], meaning $k(g\boldsymbol{x}, g'\boldsymbol{x}') = k(\boldsymbol{x}, \boldsymbol{x}')$ for all $\boldsymbol{x}, \boldsymbol{x}' \in \mathcal{S}$ and $g, g' \in \mathcal{G}$. The central question then becomes: how can one construct an invariant kernel $k$ from an arbitrary base kernel $k_\mathrm{b}$ and symmetry group $\mathcal{G}$? An elegant solution, dating back to Kondor (2008) and recently advocated for BO by Brown et al. (2024), is to average $k_\mathrm{b}$ over $\mathcal{G}$-orbits:

$$k_\mathrm{avg}(\boldsymbol{x}, \boldsymbol{x}') = \frac{1}{|\mathcal{G}|^2} \sum_{g, g' \in \mathcal{G}} k_\mathrm{b}(g\boldsymbol{x}, g'\boldsymbol{x}'). \tag{5}$$

This construction is not only guaranteed to be $\mathcal{G}$-invariant, but also admits a clean functional interpretation: if $\mathcal{H}_{k_\mathrm{b}}$ and $\mathcal{H}_{k_\mathrm{avg}}$ denote the RKHS induced by $k_\mathrm{b}$ and $k_\mathrm{avg}$ respectively, then $\mathcal{H}_{k_\mathrm{avg}}$ coincides exactly with the subspace of $\mathcal{G}$-invariant functions in $\mathcal{H}_{k_\mathrm{b}}$ (Theorem 4.4.3 in Kondor (2008)). Consequently, $k_\mathrm{avg}$ (up to normalization) has gained popularity as the standard off-the-shelf kernel for BO in symmetric settings (Glielmo et al., 2017; Kim et al., 2021; Brown et al., 2024).

A complementary idea in kernel methods is to retain the *best* latent alignment between two orbits via a maximum, as in convolution/best-match kernels for structured data (Gärtner, 2003; Vishwanathan et al., 2003) and follow-up work across domains (Fröhlich et al., 2005; Zhang, 2010; Curtin et al., 2013). Max-alignment kernels, however, are not PSD in general, leading to indefinite Gram matrices. This has motivated two families of remedies: (i) explicit Kreĭn-space formulations (Ong et al., 2004; Oglic & Gärtner, 2018), and (ii) simple PSD corrections such as eigenvalue clipping/flipping in SVMs (Luss & D'aspremont, 2007; Chen et al., 2009), which are empirically effective.

**Our adaptation to BO.** Guided by the above, we adopt the max-alignment view for BO. To ensure positive definiteness, we project $k_\mathrm{max}$ (see (1)) onto a PSD kernel $k_+^{(\mathcal{D})}$, which coincides with $k_\mathrm{max}$ whenever the latter is already PSD. This preserves the sharp, high-contrast orbit alignments of $k_\mathrm{max}$

---

[1]Up to modification, i.e., there is another GP $f'$ such that for every $x \in \mathcal{S}$, $\mathbb{P}(f(\boldsymbol{x}) = f'(\boldsymbol{x})) = 1$ and $f'$ has invariant paths and invariant covariance, see Property 3.3 in Ginsbourger et al. (2012).

while ensuring compatibility with the BO framework. Moreover, it maintains a per-iteration BO complexity comparable to that of orbit-averaged kernels (see Section 2.2). In our experiments, $k_+^{(\mathcal{D})}$ better reflects the intended symmetries of standard synthetic objectives and achieves substantially lower cumulative regret. Interestingly, these empirical gains are not mirrored by existing eigendecay-based upper bounds, a point we return to in Section 5.

# 3 THE MAX KERNEL

We have introduced the max-alignment kernel $k_{\max}$ and its PSD surrogate $k_+^{(\mathcal{D})}$ in (2). This section explains *why* $k_{\max}$ is a natural $\mathcal{G}$-invariant covariance, clarifies how it differs from orbit averaging through examples, and records the practical PSD construction we use in BO.

## 3.1 MOTIVATION: $k_{\max}$ AS A VALID COVARIANCE

A natural way to motivate $k_{\max}$ is to exhibit $\mathcal{G}$-invariant GPs whose covariance equals $k_{\max}$.

**Construction.** Let $h \sim \mathcal{GP}(0, k_\mathrm{b})$ with an isotropic base kernel $k_\mathrm{b}(\boldsymbol{x}, \boldsymbol{x}') = \kappa(\|\boldsymbol{x} - \boldsymbol{x}'\|_2)$ with $\kappa$ nonincreasing (e.g., popular ones such as RBF, Matérn). Consider a map $\phi_\mathcal{G}$ such that (i) $\phi_\mathcal{G}(\boldsymbol{x}) = \phi_\mathcal{G}(g\boldsymbol{x})$ for all $g \in \mathcal{G}$ and (ii) $\|\phi_\mathcal{G}(\boldsymbol{x}) - \phi_\mathcal{G}(\boldsymbol{x}')\|_2 = \min_{g,g'} \|g\boldsymbol{x} - g'\boldsymbol{x}'\|_2$. Define $f(\boldsymbol{x}) = h(\phi_\mathcal{G}(\boldsymbol{x}))$. Then $f$ is $\mathcal{G}$-invariant and:

**Proposition 1.** *Under the construction above, $f \sim \mathcal{GP}(0, k_{\max})$ with $k_{\max}$ given by* (1).

*Proof sketch, details in Appendix A.* $\mathrm{Cov}(f(\boldsymbol{x}), f(\boldsymbol{x}')) \stackrel{\mathrm{def}\, f}{=} k_\mathrm{b}(\phi_\mathcal{G}(\boldsymbol{x}), \phi_\mathcal{G}(\boldsymbol{x}')) \stackrel{\mathrm{(ii)}}{=} \kappa(\min_{g,g'} \|g\boldsymbol{x} - g'\boldsymbol{x}'\|_2)$, and monotonicity of $\kappa$ converts the min-distance into $\max_{g,g'} k_\mathrm{b}(g\boldsymbol{x}, g'\boldsymbol{x}')$. $\quad\square$

This shows that $k_{\max}$ naturally arises as the covariance of valid $\mathcal{G}$-invariant GPs. In contrast, the common approach to invariance in BO is to build $k_\mathrm{avg}$ by averaging a base kernel as in (5). But averaging and maximization induce fundamentally different geometries:

**Lemma 2.** *For any base kernel $k_\mathrm{b}$ and any (double) orbit $\mathcal{O}(\boldsymbol{x}, \boldsymbol{x}') := \{(g\boldsymbol{x}, g'\boldsymbol{x}'), g, g' \in \mathcal{G}\}$, $k_\mathrm{avg} = k_{\max}$ on $\mathcal{O}(\boldsymbol{x}, \boldsymbol{x}')$ if and only if $k_\mathrm{b} = k_{\max}$ on that orbit.*

Indeed, an average reaches the maximum only when every term is maximal. Thus $k_\mathrm{avg}$ can never reproduce the geometry of $k_{\max}$, except in the degenerate case where the base kernel is already $k_{\max}$, making averaging redundant. One might wonder whether this limitation of $k_\mathrm{avg}$ could be circumvented by building it from a *different* base kernel than the one used for $k_{\max}$. In Appendix A.2 we show that, under mild assumptions satisfied by standard kernels (upper-bounded by 1, with equality $k(\boldsymbol{x}, \boldsymbol{x}) = 1$ along the diagonal), $k_\mathrm{avg}$ and $k_{\max}$ can coincide only in the trivial case where the base kernel of $k_\mathrm{avg}$ is already invariant for pairs of points belonging to the same orbit. Thus, even in this more general setting, averaging does not reproduce the geometry of maximization (except if the base kernel already had invariances).

To make this contrast concrete, we now examine a simple example (radial invariance with an RBF base kernel) where $k_{\max}$ and $k_\mathrm{avg}$ can be computed in closed form.

**Example 3** (Radial invariance with $k_{\max}$)**.** *Let $\mathcal{G}$ be the group of planar rotations and $k_\mathrm{b}(\boldsymbol{x}, \boldsymbol{x}') = \exp\left(-\|\boldsymbol{x} - \boldsymbol{x}'\|_2^2/2l^2\right)$ be an RBF kernel. With $\phi_\mathcal{G}(\boldsymbol{x}) = \|\boldsymbol{x}\|_2$,*

$$k_{\max}(\boldsymbol{x}, \boldsymbol{x}') = \exp\left(-(\|\boldsymbol{x}\|_2 - \|\boldsymbol{x}'\|_2)^2/2l^2\right), \quad k_\mathrm{avg}(\boldsymbol{x}, \boldsymbol{x}') = \exp\left(-\frac{\|\boldsymbol{x}\|_2^2 + \|\boldsymbol{x}'\|_2^2}{2l^2}\right) I_0\left(\frac{\|\boldsymbol{x}\|_2 \|\boldsymbol{x}'\|_2}{l^2}\right),$$

*with $I_0$ the modified Bessel function (derivation in Appendix B). As illustrated in Figure 1, the two kernels $k_{\max}$ and $k_\mathrm{avg}$ induce qualitatively different similarity structures. By construction, $k_{\max}$ assigns large similarity whenever $\|\boldsymbol{x}\|_2 \approx \|\boldsymbol{x}'\|_2$. If $\|\boldsymbol{x}\|_2 = \|\boldsymbol{x}'\|_2$, the function $f^\star$ satisfies $f^\star(\boldsymbol{x}) = f^\star(\boldsymbol{x}')$ since it is invariant under rotations, and $k_{\max}$ exactly recovers this invariance by assigning maximal similarity $k_{\max}(\boldsymbol{x}, \boldsymbol{x}') = 1$. In contrast, $k_\mathrm{avg}$ only approximates this behavior: its iso-similarity curves as a function of $(\|\boldsymbol{x}\|_2, \|\boldsymbol{x}'\|_2)$ correspond to distorted balls, and two points with identical norms may be ranked as highly dissimilar (see the diagonal $\|\boldsymbol{x}\|_2 = \|\boldsymbol{x}'\|_2$ of the right plot in Figure 1). This mismatch highlights that while both constructions enforce rotation invariance, only $k_{\max}$ preserves the correct notion of similarity.*

Table 1: Complexity per BO iteration. Here $|G|^*$ denotes either $|G|$ or $|G|^2$ depending on whether the orbit terms reduce to a single sum (when $k_\mathrm{b}(g\boldsymbol{x}, \boldsymbol{x}')$ suffices) or require a double sum over $(g, g')$; $m$ is the number of candidate points used in acquisition optimization. The row *Per-candidate acquisition evaluation* gives the cost of a single acquisition evaluation; for one BO iteration this row is multiplied by $m$ and added to the other rows to obtain the total.

|  | Base kernel $k_\mathrm{b}$ | Averaged $k_\mathrm{avg}$ | Projected $k_+^{(\mathcal{D})}$ |
|---|---|---|---|
| Gram matrix ($n \times n$) | $\mathcal{O}(n^2)$ | $\mathcal{O}(n^2|G|^*)$ | $\mathcal{O}(n^2|G|^*)$ |
| SVD / inversion | $\mathcal{O}(n^3)$ | $\mathcal{O}(n^3)$ | $\mathcal{O}(n^3)$ |
| PSD projection | – | – | $\mathcal{O}(n^3)^4$ |
| Per-candidate acq. eval. | $\mathcal{O}(1)$ | $\mathcal{O}(|G|^*)$ | $\mathcal{O}(n|G|^*)$ |
| **Total for 1 BO iteration** | $\mathcal{O}(m + n^2 + n^3)$ | $\mathcal{O}((m + n^2)|G|^* + n^3)$ | $\mathcal{O}((mn + n^2)|G|^* + n^3)$ |

## 3.2 A PSD Extension of $k_{\max}$: What We Use in Practice

Because $k_{\max}$ is not PSD in general, we apply a standard projection step on the finite design set $\mathcal{D} = \{\boldsymbol{x}_1, \ldots, \boldsymbol{x}_n\}$. Let $\boldsymbol{K} = k_{\max}(\mathcal{D}, \mathcal{D})$ with eigendecomposition $\boldsymbol{K} = \boldsymbol{Q}\boldsymbol{\Lambda}\boldsymbol{Q}^\top$ and define[2] (with the max applied elementwise)

$$\boldsymbol{K}_+ = \boldsymbol{Q} \max(0, \boldsymbol{\Lambda}) \boldsymbol{Q}^\top. \tag{6}$$

We then use the Nyström extension[3] (Williams & Seeger, 2000) to evaluate cross-covariances with new points, yielding the PSD, $\mathcal{G}$-invariant surrogate $k_+^{(\mathcal{D})}$ given in (2) and that we reproduce here:

$$k_+^{(\mathcal{D})}(\boldsymbol{x}, \boldsymbol{x}') := k_{\max}(\boldsymbol{x}, \mathcal{D}) \, \boldsymbol{K}_+^\dagger \, k_{\max}(\mathcal{D}, \boldsymbol{x}'). \tag{7}$$

**Key properties of $k_+^{(D)}$:**

- *PSD & invariance.* $k_+^{(\mathcal{D})}$ is PSD and inherits argumentwise $\mathcal{G}$-invariance[5] of $k_{\max}$.
- *Consistency with $k_{\max}$.* If $\boldsymbol{K} \succeq 0$, then $\boldsymbol{K}_+ = \boldsymbol{K}$ and $k_+^{(\mathcal{D})}$ agrees with $k_{\max}$ on $\mathcal{D} \times \mathcal{D}$.
- *Cost.* Each BO iteration involves (i) building the Gram matrix on $\mathcal{D}$, (ii) inverting the Gram matrix to build the acquisition function, and (iii) $m$ kernel evaluations when optimizing the acquisition function. Step (ii) has the same cost as the SVD of $\boldsymbol{K}$ needed to compute both $\boldsymbol{K}_+$ and $\boldsymbol{K}_+^\dagger$, which makes $k_+^{(\mathcal{D})}$ having the same asymptotic per-iteration cost as $k_{\max}$; its per-query evaluations are more expensive, but this difference is negligible as long as we keep $m \lesssim n$. A concise complexity summary is provided in Table 1, and example of runtimes in Table 3.
- *Regularity.* For finite groups, $k_{\max}$ is a max of finitely many smooth maps and is almost everywhere (a.e.) differentiable; the Nyström extension preserves a.e. differentiability in each argument. For continuous groups, smoothness can sometimes be obtained via closed-form formulas (e.g., as in Example 3).

We now illustrate the behavior of $k_+^{(\mathcal{D})}$ versus $k_{\max}$ (in this situation, $k_{\max}$ is not PSD and the projection step is indeed needed to restore positive semidefiniteness).

**Example 4** (Ackley function with $k_+$). *Figure 2 compares $k_+^{(\mathcal{D})}$ and $k_{\max}$ on the one-dimensional Ackley function (see (15)). The projected kernel $k_+^{(\mathcal{D})}$ preserves the expected pairwise symmetries (invariance along $x = y$ and $x = -y$) and spreads mass more evenly across the symmetric regions, whereas $k_{\max}$ concentrates covariance mostly near the origin. Thus, $k_+^{(\mathcal{D})}$ better reflects the symmetry geometry of the problem, echoing the qualitative difference observed in Example 3.*

**Beyond the finite view (details in Appendix C).** The PSD projection with Nyström in Equation (7) is a practical, data-dependent construction. It can be seen as the finite-sample face of a broader, intrinsic

---

[2]$\boldsymbol{K}_+$ does not depend on the choice of the eigendecomposition, see Lemma 7 in the appendix.

[3]It indeed extends $\boldsymbol{K}_+$ since $k_+^{(\mathcal{D})}(\boldsymbol{x}_i, \boldsymbol{x}_j) = \boldsymbol{K}_{i,:} \, \boldsymbol{K}_+^\dagger \, \boldsymbol{K}_{:,j} = (\boldsymbol{K}\boldsymbol{K}_+^\dagger\boldsymbol{K})_{ij} = (\boldsymbol{K}_+)_{ij}$.

[4]One SVD of $\boldsymbol{K}$ suffices to obtain both $\boldsymbol{K}_+$ and $\boldsymbol{K}_+^\dagger$, so the extra PSD projection does not increase asymptotic cost.

[5]$k_{\max}(g\boldsymbol{x}, \boldsymbol{x}') = k_{\max}(\boldsymbol{x}, \boldsymbol{x}')$ implies $k_{\max}(g\boldsymbol{x}, \mathcal{D}) = k_{\max}(\boldsymbol{x}, \mathcal{D})$, hence invariance of $k_+^{(\mathcal{D})}$.

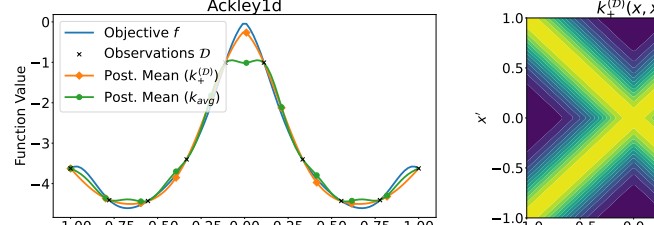 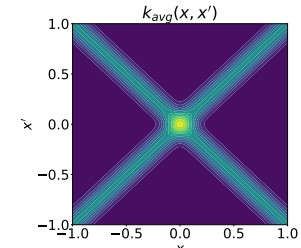

Figure 2: (Left) One-dimensional Ackley function $f^\star$ (see (15)), invariant up to coordinate-wise sign-flips, and GP posterior means $\mu_t(\boldsymbol{x})$ as in (3) for $k_+^{(\mathcal{D})}$ (orange diamond) and $k_{\text{avg}}$ (green circles) built from $\mathcal{D}$ (black crosses). (Center) Covariance structure induced by $k_+^{(\mathcal{D})}$. (Right) Covariance structure induced by $k_{\text{avg}}$. Both kernels are invariant to reflections across $x = y$ and $x = -y$, but $k_{\text{avg}}$ concentrates covariance near 0, while $k_+^{(\mathcal{D})}$ better reflects the underlying symmetry geometry. Consequently, the GP posterior mean induced by $k_+^{(\mathcal{D})}$ is the best at fitting the objective (left).

Table 2: Performance of $k_{\text{b}}$, $k_{\text{avg}}$, and $k_+^{(\mathcal{D})}$ across benchmarks. For each kernel $k \in \{k_{\text{b}}, k_{\text{avg}}, k_+^{(\mathcal{D})}\}$ we report $m \pm s_{\text{err}}$, where $m$ is the empirical mean over 10 seeds (lower is better) and $s_{\text{err}}$ is the empirical standard error. Best mean is **bold**; means $m$ whose 95% confidence interval ($m \pm 1.96 s_{\text{err}}$) confidence interval overlap with the best are underlined. Performance is measured by cumulative regret on synthetic benchmarks and by negated simple reward on real-world experiments.

| **Benchmark** | $|\mathcal{G}|$ | $k_{\text{b}}$ | $k_{\text{avg}}$ | $k_+^{(\mathcal{D})}$ |
|---|---|---|---|---|
| *Synthetic (Cumulative Reg.)* | | | | |
| Ackley2d | 8 | $382.7 \pm 5.7$ | $128.2 \pm 10.4$ | $\mathbf{126.4 \pm 3.6}$ |
| Griewank6d | 64 | $3840.3 \pm 177.7$ | $3067.4 \pm 841.9$ | $\mathbf{1832.6 \pm 146.3}$ |
| Rastrigin5d | $3,840$ | $3568.5 \pm 91.3$ | $1583.5 \pm 341.9$ | $\mathbf{813.4 \pm 70.6}$ |
| Radial2d | $\infty$ | $388.6 \pm 20.3$ | $480.9 \pm 76.4$ | $\mathbf{199.7 \pm 11.6}$ |
| Scaling2d | $\infty$ | $1820.6 \pm 1135.4$ | $3361.8 \pm 742.9$ | $\mathbf{25.4 \pm 6.4}$ |
| *Real-World (Neg. Simple Rew.)* | | | | |
| WLAN8d | 24 | $-65.0 \pm 3.2$ | $-51.8 \pm 1.7$ | $\mathbf{-74.4 \pm 0.7}$ |
| PartPack6d | $\infty$ | $-0.79 \pm 0.10$ | $-0.69 \pm 0.01$ | $\mathbf{-0.92 \pm 0.10}$ |

definition that does not depend on $\mathcal{D}$. Since $k_{\max}$ is symmetric, it admits a spectral decomposition $k_{\max}(\boldsymbol{x}, \boldsymbol{x}') = \sum_i \lambda_i \phi_i(\boldsymbol{x}) \phi_i(\boldsymbol{x}')$ in $L^2$, and we can always define (a.e.)

$$k_+(\boldsymbol{x}, \boldsymbol{x}') := \sum_i \max(0, \lambda_i)\, \phi_i(\boldsymbol{x}) \phi_i(\boldsymbol{x}'),$$

with $k_+ = k_{\max}$ whenever $k_{\max}$ is already PSD. On finite domains, this precisely reduces to the matrix PSD projection in (6). In Appendix C we formalize the infinite-domain construction via integral operators, prove that $k_+$ is $\mathcal{G}$-invariant, and show that the finite projection + Nyström in (7) converges to $k_+$ at the spectral (Hilbert-Schmidt) level under iid sampling (Appendix C.4).

**Takeaway.** $k_{\max}$ is the exact covariance of a natural class of $\mathcal{G}$-invariant GPs and induces a search geometry that preserves high-contrast orbit alignments (Examples 3 and 4). The PSD projection + Nyström step yields a valid GP kernel $k_+^{(\mathcal{D})}$ without introducing extra asymptotic complexity. We now measure its practical impact in Section 4.

## 4 EXPERIMENTS

We evaluate $k_+^{(\mathcal{D})}$ against two baselines: (i) the off-the-shelf kernel $k_{\text{b}}$ (no symmetry handling), and (ii) the orbit-averaged kernel $k_{\text{avg}}$ (Brown et al., 2024). Benchmarks include standard synthetic objectives and two real-world tasks with known invariances (a wireless network design task and a particle packing problem). We ask: *(Q1) Does $k_+^{(\mathcal{D})}$ reduce simple/cumulative regret vs. $k_{\text{avg}}$?* and *(Q2) How does performance scale with the size of the symmetry group and dimension?* The full experimental setup is described in Appendix E.

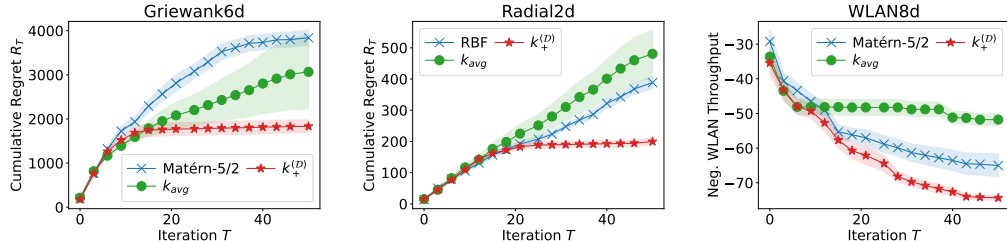

Figure 3: Cumulative regret and negated simple reward under GP-UCB with $k_{\mathrm{b}}$ (blue crosses), $k_{\mathrm{avg}}$ (orange diamonds), and $k_+^{(\mathcal{D})}$ (green circles) on a selection of benchmarks (all benchmarks in Appendix E). Shaded regions show the standard error ($\pm s_{err}$) over 10 seeds.

**Headline: $k_+^{(\mathcal{D})}$ wins on every task.** Across all benchmarks (Table 2), $k_+^{(\mathcal{D})}$ achieves the best performance with up to 50% of improvement. This answers **Q1** positively. Regarding **Q2**, we will see that as the group size increases, $k_+^{(\mathcal{D})}$ stays strong, while $k_{\mathrm{avg}}$ degrades and can even underperform the non-invariant base kernel $k_{\mathrm{b}}$.

**Setup in one glance.** We run GP-UCB with each kernel $k \in \{k_{\mathrm{b}}, k_{\mathrm{avg}}, k_+^{(\mathcal{D})}\}$, using the same acquisition and optimization budgets. We report results averaged over 10 seeds. All the hyperparameters and group actions are detailed in Appendix E.

### 4.1 SYNTHETIC BENCHMARKS

We consider synthetic functions $f^\star$ (Ackley, Griewank, Rastrigin, etc.) that exhibit symmetries (such as permutations, coordinate-wise sign-flips, rotations, rescaling) and are classically considered as challenging to optimize in the BO literature (Qian et al., 2021; Bardou et al., 2024). We cover dimensions $d = 2$ to $d = 6$ and group sizes $|\mathcal{G}| = 8$ to $|\mathcal{G}| = \infty$. We evaluate performance using the cumulative regret $R_T = \sum_{i=1}^{T} \left( f^\star(\boldsymbol{x}^*) - f^\star(\boldsymbol{x}_t) \right)$ since the global maximizer $\boldsymbol{x}^* = \arg\max_{\boldsymbol{x} \in \mathcal{S}} f^\star(\boldsymbol{x})$ is known.

**Finite groups: the gap widens as $|\mathcal{G}|$ grows.** With Matérn-5/2 base $k_{\mathrm{b}}$ on Ackley2d ($|\mathcal{G}|$=8), $k_{\mathrm{avg}}$ and $k_+^{(\mathcal{D})}$ are tied; both dominate $k_{\mathrm{b}}$. As $|\mathcal{G}|$ increases (Griewank6d, $|\mathcal{G}|$=64; Rastrigin5d, $|\mathcal{G}|$=3,840), $k_+^{(\mathcal{D})}$ increasingly outperforms $k_{\mathrm{avg}}$ achieving cumulative regrets that are, on average, 40% and 49% lower respectively (Table 2, Figure 3 left panel, and Appendix E for the whole set of figures).

**Continuous groups: $k_{\mathrm{avg}}$ can underperform even $k_{\mathrm{b}}$.** For radial and scaling invariances (continuous groups; RBF base), $k_{\mathrm{avg}}$ degrades relative to $k_{\mathrm{b}}$, while $k_+^{(\mathcal{D})}$ remains strong (Figure 3 center panel, and Appendix E for the whole set of figures).

### 4.2 REAL-WORLD EXPERIMENTS

We consider two real-world experiments that are described in detail in Appendix E: the design of a wireless network (8-dimensional, invariant to permutations of pairs of parameters) and a particle packing problem (6-dimensional, invariant to the rescaling of some parameters and to permutations of pairs of parameters). For both benchmarks, performance is evaluated using the negated best reward $\min_{t \in [T]} -f^\star(\boldsymbol{x}_t)$ attained during optimization (the regret cannot be computed because the max of $f^\star$ is unknown). Note that we consider $\min_{t \in [T]} -f^\star(\boldsymbol{x}_t)$ instead of the cumulated $-\sum_t f^\star(\boldsymbol{x}_t)$ because the goal is to assess the quality of the best combination of parameters discovered by the optimizer, rather than the cumulative negative reward across all explored combinations.

$k_+^{(\mathcal{D})}$ **finds better combinations of parameters.** For the design of a wireless network or for the particle packing problem, $k_+^{(\mathcal{D})}$ consistently discovers combinations of parameters with larger utility than both $k_{\mathrm{avg}}$ and $k_{\mathrm{b}}$ (Figure 3 right; Appendix E for more figures).

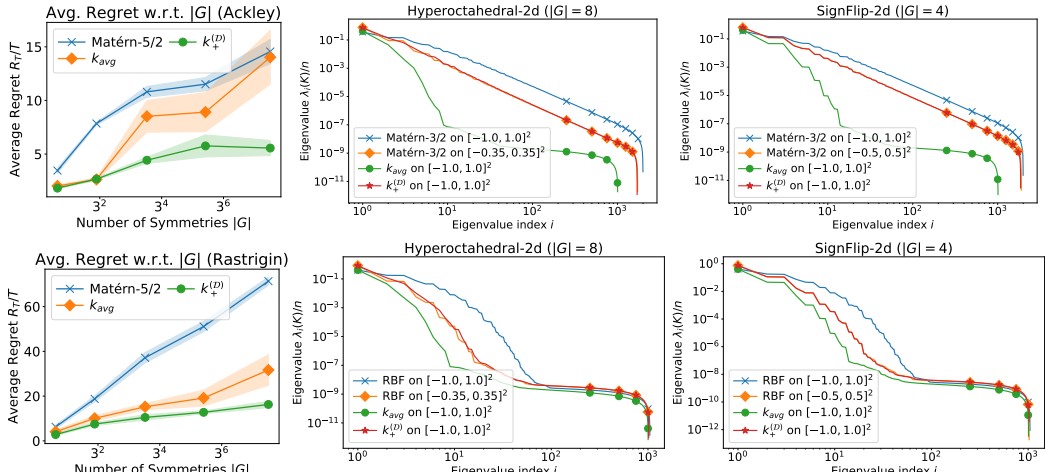

Figure 4: **Left column:** Final average regret $R_T/T$ for $k_{\mathrm{b}}$ (blue crosses), $k_{\mathrm{avg}}$ (orange diamonds), and $k_+^{(\mathcal{D})}$ (green circles) on Ackley (top) and Rastrigin (bottom), averaged over 10 seeds with standard error bars. **Middle and right columns:** Empirical eigendecays under different bases and groups (ordered eigenvalues of the Gram-matrix divided by $n$), typical behavior on a single seed.

## 4.3 ROBUSTNESS TO GROUP SIZE

Both synthetic and real-world benchmarks suggest that $k_{\mathrm{avg}}$ performs comparably to $k_+^{(\mathcal{D})}$ when the group size $|\mathcal{G}|$ is small, but its performance deteriorates as $|\mathcal{G}|$ grows, whereas $k_+^{(\mathcal{D})}$ remains stable. To investigate this effect more systematically, we conduct additional experiments on the $d$-dimensional Ackley and Rastrigin benchmarks, each invariant under the hyperoctahedral group $\mathcal{G}$ of size $|\mathcal{G}| = 2^d d!$ (permutations × coordinate-wise sign-flips). We compare the average regret of $k_{\mathrm{avg}}$ and $k_+^{(\mathcal{D})}$ after 50 iterations of GP-UCB for dimensions $d = 1, \ldots, 5$, and include $k_{\mathrm{b}}$ as a baseline to control for the effect of increasing $d$.

The results are shown in Figure 4 (left column) . Both experiments reveal the same trend: while $k_{\mathrm{avg}}$ consistently outperforms $k_{\mathrm{b}}$, its performance also deteriorates as $|\mathcal{G}|$ increases. In contrast, $k_+^{(\mathcal{D})}$ remains largely unaffected by the growing number of symmetries, demonstrating a clear robustness to group size. In the next section, we discuss several explanations for these empirical observations.

**Takeaway.** $k_+^{(\mathcal{D})}$ consistently matches or outperforms $k_{\mathrm{avg}}$ and $k_{\mathrm{b}}$, with the largest gains at large $|\mathcal{G}|$. The evidence suggests that (i) *how* a kernel encodes orbit alignments matters as much as *whether* it is invariant, and (ii) averaging across many alignments can dilute informative similarities. These themes reconnect with our discussion in Section 5 and motivate analyses beyond eigendecay rates.

## 5 SPECTRAL ANALYSIS AND REGRET BOUNDS

So far, $k_+^{(\mathcal{D})}$ has shown consistently lower regret than $k_{\mathrm{avg}}$, despite comparable computational cost. A natural question is: *can existing BO theory account for such a gap?* Current regret bounds for GP surrogates proceed via the information gain, which is shaped by the decay of the operator spectrum of the kernel. In particular, faster spectral decay leads to tighter regret upper bounds in standard analyses (Srinivas et al., 2012; Valko et al., 2013; Scarlett et al., 2017; Whitehouse et al., 2023). We now compare the eigendecay of $k_+^{(\mathcal{D})}$ and $k_{\mathrm{avg}}$, and ask whether it can explain the empirical gap.

**Empirical eigendecays: similar or *faster* decay for $k_{\mathrm{avg}}$.** Across our benchmarks, the empirical spectra of $k_+^{(\mathcal{D})}$ and $k_{\mathrm{avg}}$ exhibit very similar log–log slopes (decay rates). In several settings, $k_{\mathrm{avg}}$'s eigenvalues decay even *faster* than those of $k_+$; see Figure 4 (middle and right columns). Under the usual theory, this would translate into similar, or potentially *tighter*, upper bounds for methods run with $k_{\mathrm{avg}}$ compared to those with $k_+^{(\mathcal{D})}$. A more detailed discussion of the empirical spectra in Figure 4 and further insights are in Appendix D.

**Limitations of eigendecay as an explanation.** Since $k_{\mathrm{avg}}$ matches or exceeds $k_+^{(\mathcal{D})}$ in empirical decay rate, standard theory would predict similar or better regret upper bounds. Yet in practice we consistently observe lower regret for $k_+^{(\mathcal{D})}$ (Section 4). This suggests that eigendecay alone does not capture the structural advantages of $k_+^{(\mathcal{D})}$. We outline possible explanations in the conclusion.

## 6   Conclusion

Our spectral analysis highlights a gap between theory and practice: although $k_{\mathrm{avg}}$ often exhibits *faster* empirical eigendecay than $k_+^{(\mathcal{D})}$, the latter consistently achieves lower regret. Standard eigendecay arguments thus fail to explain the observed advantage of $k_+^{(\mathcal{D})}$. We hypothesize two complementary explanations.

First, **geometry vs. rates:** eigendecay quantifies how fast spectra shrink but ignores *which* eigenfunctions are emphasized. In practice, $k_{\mathrm{avg}}$ often introduces *similarity reversals*, distorting the search geometry (Figure 1), whereas $k_+^{(\mathcal{D})}$ preserves high-contrast alignments between orbits, just as $k_{\max}$.

Second, **approximation hardness:** standard BO guarantees assume that the black-box $f^\star$ belongs to the RKHS $\mathcal{H}_k$ of the chosen kernel $k$. In practice, this assumption is unverifiable. One might therefore wonder whether the empirical gap between $k_{\mathrm{avg}}$ and $k_+^{(\mathcal{D})}$ could stem from *misspecification*, i.e., different distances $d(f^\star, \mathcal{H}_{k_{\mathrm{avg}}})$ and $d(f^\star, \mathcal{H}_{k_+^{(\mathcal{D})}})$. Indeed, existing work (Bogunovic & Krause, 2021) shows that when $d(f^\star, \mathcal{H}_k) > 0$, regret necessarily degrades linearly with this distance.

However, this explanation does not apply in our setting. With an RBF base kernel $k_{\mathrm{b}}$, $\mathcal{H}_{k_{\mathrm{b}}}$ is universal (Micchelli et al., 2006), so $\mathcal{H}_{k_{\mathrm{avg}}}$ contains all invariant functions[6] including $f^\star$, so $d(f^\star, \mathcal{H}_{k_{\mathrm{avg}}}) = 0$ and cannot exceed $d(f^\star, \mathcal{H}_{k_+^{(\mathcal{D})}})$. Yet $k_{\mathrm{avg}}$ still performs worse than $k_+^{(\mathcal{D})}$.

This suggests that the relevant issue is not membership in $\mathcal{H}_{k_{\mathrm{avg}}}$ or $\mathcal{H}_{k_+^{(\mathcal{D})}}$, but *how difficult $f^\star$ is to approximate within each RKHS*. BO ultimately constructs approximations from finite linear combinations of atoms $k(\boldsymbol{x}_t, \cdot)$. Different kernels can thus induce different approximation rates.

This perspective might explain both our results and those of Brown et al. (2024). In their setup, the synthethic benchmarks $f^\star$ are explicit linear combinations of relatively few $k_{\mathrm{avg}}(\boldsymbol{x}_t, \cdot)$ atoms (between 64 and 512, depending on dimension; see their Appendix B.1). In such regimes, $k_{\mathrm{avg}}$ performs very well, likely because the GP posterior mean can in principle exactly recover $f^\star$ once those $\boldsymbol{x}_t$ are sampled. Typical BO objectives do not exhibit this sparse expansion structure, which may explain why in our experiments $k_{\mathrm{avg}}$ consistently underperforms $k_+^{(\mathcal{D})}$, and sometimes even the base kernel.

Developing regret bounds that capture this notion of approximation hardness, beyond mere membership in $\mathcal{H}_k$, appears essential to bridge the gap between theory and empirical performance.

Finally, while our focus has been empirical, we note that the intrinsic data-independent version of $k_+^{(\mathcal{D})}$, which we called $k_+$ and which we mentioned at the end of Section 3.2 (introduced formally in Appendix C), provides a natural, data-independent analogue of the practical kernel $k_+^{(\mathcal{D})}$. We see $k_+$ as a convenient object for future theoretical work, as it cleanly isolates the PSD projection of $k_{\max}$ from the additional data dependence introduced by Nyström. We believe that it makes $k_+$ a convenient starting point for any future theoretical work, in the same spirit as gradient flow serving as an idealized analogue of gradient descent.

### Acknowledgments

We thank the anonymous reviewers for their constructive suggestions, which led to several improvements and clarifications in the final version. This work was supported in part by the Swiss State Secretariat for Education, Research and Innovation (SERI) under contract number MB22.00027.

---

[6] Consider $(Pf)(\boldsymbol{x}) = \sum_{g \in \mathcal{G}} f(g\boldsymbol{x})/|\mathcal{G}|$, the projection onto $\mathcal{H}_{k_{\mathrm{avg}}}$ (Brown et al., 2024, Appendix A), and observe that if $f_n \to f^\star$ with $f_n \in \mathcal{H}_{k_{\mathrm{b}}}$ then $Pf_n \to f^\star$ with $Pf_n \in \mathcal{H}_{k_{\mathrm{avg}}}$. So universality of $\mathcal{H}_{k_{\mathrm{b}}}$ implies universality of $\mathcal{H}_{k_{\mathrm{avg}}}$.

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

# A PROOFS FOR SECTION 3

## A.1 FULL STATEMENT AND PROOF OF PROPOSITION 1

We state Proposition 1 formally and give a slightly more detailed proof.

**Proposition 5** (Max-kernel covariance for invariant GPs). *Let $\mathcal{S}, \mathcal{S}_h \subset \mathbb{R}^d$ be measurable spaces and let a (finite or compact) group $\mathcal{G}$ act measurably on $\mathcal{S}$. Let $h \sim \mathcal{GP}(0, k_{\mathrm{b}})$ be a GP on $\mathcal{S}_h$ with an isotropic base kernel $k_{\mathrm{b}} : (\boldsymbol{x}, \boldsymbol{x}') \in \mathcal{S} \times \mathcal{S} \mapsto \kappa(\|\boldsymbol{x} - \boldsymbol{x}'\|_2)$ where $\kappa : \mathbb{R}_{\geq 0} \to \mathbb{R}_{\geq 0}$ is nonincreasing. Assume there exists $\phi_{\mathcal{G}} : \mathcal{S} \to \mathcal{S}_h$ satisfying (i)* invariance: *$\phi_{\mathcal{G}}(\boldsymbol{x}) = \phi_{\mathcal{G}}(g\boldsymbol{x})$ for all $g \in \mathcal{G}, \boldsymbol{x} \in \mathcal{S}$; and (ii)* minimal-distance representativity: *$\|\phi_{\mathcal{G}}(\boldsymbol{x}) - \phi_{\mathcal{G}}(\boldsymbol{x}')\|_2 = \min_{g, g' \in \mathcal{G}} \|g\boldsymbol{x} - g'\boldsymbol{x}'\|_2$. Define $f(\boldsymbol{x}) = h(\phi_{\mathcal{G}}(\boldsymbol{x}))$. Then $f \sim \mathcal{GP}(0, k_{\max})$ and it is $\mathcal{G}$-invariant.*

*Proof.* Since $g$ is a GP, $f$ is also a GP, and invariance follows from (i). Its covariance kernel is $k_{\max}$ since:

$$\mathrm{Cov}\left[f(\boldsymbol{x}), f(\boldsymbol{x}')\right] = \mathrm{Cov}\left[h(\phi_{\mathcal{G}}(\boldsymbol{x})), h(\phi_{\mathcal{G}}(\boldsymbol{x}'))\right]$$

$$= k_{\mathrm{b}}(\phi_{\mathcal{G}}(\boldsymbol{x}), \phi_{\mathcal{G}}(\boldsymbol{x}'))$$

$$= \kappa(\min_{g, g' \in \mathcal{G}} \|g\boldsymbol{x} - g'\boldsymbol{x}'\|_2) \tag{8}$$

$$= \max_{g, g' \in \mathcal{G}} \kappa(\|g\boldsymbol{x} - g'\boldsymbol{x}'\|_2) \tag{9}$$

$$= k_{\max}(\boldsymbol{x}, \boldsymbol{x}') \tag{10}$$

where we used (ii) in Equation (8), and monotonicity of $\kappa$ in Equation (9). Note that compactness of $G$ guarantees that the minimum in (ii) is indeed achieved, which makes Equation (9) true even when $\kappa$ is not necessarily continuous. $\square$

## A.2 AVERAGING VS MAXIMIZATION WITH DIFFERENT BASE KERNELS

We extend Lemma 2 to the case where $k_{\mathrm{avg}}$ and $k_{\max}$ are built from *different* base kernels. The result shows that even in this more flexible setting, the coincidence of $k_{\mathrm{avg}}$ and $k_{\max}$ can only occur in degenerate situations.

**Lemma 6.** *Let $k_{\mathrm{b}}$ and $k'_{\mathrm{b}}$ be two base kernels such that $\|k_{\mathrm{b}}\|_{\infty} = \|k'_{\mathrm{b}}\|_{\infty} = 1$ and $k'_{\mathrm{b}}(\boldsymbol{x}, \boldsymbol{x}) = 1$ for all $\boldsymbol{x}$. Let $k_{\mathrm{avg}}$ be the group-averaged kernel built from $k_{\mathrm{b}}$ and $k_{\max}$ be the maximization kernel built from $k'_{\mathrm{b}}$. It holds*

$$k_{\mathrm{avg}} = k_{\max} \quad \text{on the orbit } \mathcal{O}(\boldsymbol{x}, g\boldsymbol{x}) := \{(h\boldsymbol{x}, h'g\boldsymbol{x}), \ h, h' \in \mathcal{G}\}$$

*for every $\boldsymbol{x} \in \mathcal{X}$ and $g \in \mathcal{G}$, if and only if*

$$k_{\mathrm{b}}(\boldsymbol{x}, g\boldsymbol{x}) = k_{\max}(\boldsymbol{x}, g\boldsymbol{x}) = 1 \quad \text{for every } \boldsymbol{x} \text{ and } g \in \mathcal{G}.$$

*In particular, this forces $k_{\mathrm{b}}$ to already exhibit a form of $\mathcal{G}$-invariance on pairs $(\boldsymbol{x}, g\boldsymbol{x})$.*

*Proof.* ($\Rightarrow$) Fix $\boldsymbol{x}$ and $g \in \mathcal{G}$. Since by assumption $k'_{\mathrm{b}}$ is bounded by 1 and $k'_{\mathrm{b}}(\boldsymbol{x}, \boldsymbol{x}) = 1$:

$$1 \geq k_{\max}(\boldsymbol{x}, g\boldsymbol{x}) = \max_{h, h' \in \mathcal{G}} k'_{\mathrm{b}}(h\boldsymbol{x}, h'g\boldsymbol{x}) \geq k'_{\mathrm{b}}(\boldsymbol{x}, \boldsymbol{x}) = 1$$

so $k_{\max}(\boldsymbol{x}, g\boldsymbol{x}) = 1$.

Now consider $k_{\mathrm{avg}}$. By definition,

$$k_{\mathrm{avg}}(\boldsymbol{x}, g\boldsymbol{x}) = \frac{1}{|\mathcal{G}|^2} \sum_{h, h' \in \mathcal{G}} k_{\mathrm{b}}(h\boldsymbol{x}, h'g\boldsymbol{x}).$$

Each summand is bounded by 1 and the average is equal to 1 as $k_{\mathrm{avg}}(\boldsymbol{x}, g\boldsymbol{x}) = k_{\max}(\boldsymbol{x}, g\boldsymbol{x}) = 1$. Therefore each term is equal to 1, which proves $k_{\mathrm{b}} = k_{\max} = 1$ on $\mathcal{O}(\boldsymbol{x}, g\boldsymbol{x})$. As this is true for every $\boldsymbol{x}, g \in \mathcal{G}$, this shows the result. The converse is immediate. $\square$

This shows that even when allowing different base kernels for $k_{\mathrm{avg}}$ and $k_{\max}$, equality between the two kernels requires $k_{\mathrm{b}}$ to already be argumentwise $\mathcal{G}$-invariant on pairs $(\boldsymbol{x}, g\boldsymbol{x})$. This fails for standard choices (e.g. RBF kernels with translation or rotation groups), so averaging cannot replicate maximization in practice.

## B  RADIAL INVARIANCE: CLOSED FORM FOR $k_{\mathrm{avg}}$

We prove the formulas provided in Example 3. Let $\mathcal{G} = \mathrm{SO}(2)$ act on $\mathbb{R}^2$ by in-plane rotations, and let $k_{\mathrm{b}}$ be the RBF kernel with lengthscale $l$: $k_{\mathrm{b}}(\boldsymbol{x}, \boldsymbol{x}') = \exp\big(-\|\boldsymbol{x} - \boldsymbol{x}'\|_2^2/(2l^2)\big)$. Writing $\boldsymbol{x} = (r, \theta)$ and $\boldsymbol{x}' = (s, \varphi)$ in polar coordinates, we have

$$k_{\mathrm{avg}}(\boldsymbol{x}, \boldsymbol{x}') = \frac{1}{(2\pi)^2} \int_0^{2\pi} \int_0^{2\pi} \exp\Big( - \tfrac{r^2 + s^2 - 2rs\cos(\theta - \varphi + \alpha - \beta)}{2l^2} \Big) \, d\alpha \, d\beta.$$

Integrating out the absolute angle and keeping only the relative angle $\psi = \theta - \varphi + \alpha - \beta$ yields

$$k_{\mathrm{avg}}(\boldsymbol{x}, \boldsymbol{x}') = \exp\Big(-\tfrac{r^2 + s^2}{2l^2}\Big) \cdot \frac{1}{2\pi} \int_0^{2\pi} \exp\big(\tfrac{rs}{l^2} \cos\psi\big) \, d\psi = \exp\Big(-\tfrac{r^2 + s^2}{2l^2}\Big) I_0\big(\tfrac{rs}{l^2}\big),$$

where $I_0(z) = \frac{1}{2\pi} \int_0^{2\pi} e^{z\cos\psi} \, d\psi$ is the modified Bessel function of order 0.

## C  AN INTRINSIC PSD PROJECTION $k_+$ AND ITS PROPERTIES

We first recall in Appendix C.1 why positive semidefiniteness is essential in the BO framework. Then, we turn to the intrinsic structure underlying our practical construction.

In the main text we defined a *data-dependent* kernel $k_+^{(\mathcal{D})}$, obtained by projecting the Gram matrix of $k_{\mathrm{max}}$ on a finite set of samples $\mathcal{D}$ onto the PSD cone and extending it via Nyström. This finite-sample construction $k_+^{(\mathcal{D})}$ is the star of the show in practice, as it is convenient to compute and exhibits strong empirical performance. However, its data-dependence can complicate theoretical analysis.

In this appendix, we show that $k_+^{(\mathcal{D})}$ is the finite-sample facet of a broader, intrinsic *data-independent* PSD projection $k_+$ of $k_{\mathrm{max}}$ which (i) preserves the $\mathcal{G}$-invariance of $k_{\mathrm{max}}$, (ii) coincides with $k_{\mathrm{max}}$ whenever $k_{\mathrm{max}}$ is already PSD. Since the PSD projection of $k_{\mathrm{max}}$ discussed here can also be applied to any other indefinite kernel $k$, we directly introduce it for an arbitrary kernel $k$.

We present the finite-domain "matrix" construction to build intuition in Appendix C.2, and then lift it to general domains via integral operators.

### C.1  WHY PSDNESS OF $k$ MATTERS

In this paper, we consider $k = k_{\mathrm{max}}$ and then project it onto a PSD kernel. Although there is *no technical impossibility* in running a BO loop as in Section 2.1 with a kernel $k$ that is not PSD,[7] doing so is poorly motivated: the fundamental assumptions underlying BO no longer apply, and the key quantities lose their meaning. In particular:

- the assumption $f^\star \in \mathcal{H}_k$ no longer makes sense because $\mathcal{H}_k$ is not defined for non-PSD kernels;
- the usual interpretation of BO as maintaining a GP prior whose posteriors provide increasingly refined approximations of $f^\star$ no longer holds (in particular $\mu_t$ and $\mathrm{Cov}_t$ are no longer GP posterior mean or covariance), since $k$ is not a valid covariance structure for the prior;
- acquisition functions (UCB, EI, etc.) lose their principled exploration-exploitation meaning and may now behave unpredictably.

### C.2  WARMUP: FINITE DOMAINS

We start on a finite domain $\mathcal{S}$ to build intuition. In that case, $k_+$ is simply Frobenius-nearest PSD truncation of the Gram matrix on the *full domain* $\mathcal{S}$, which is unique, basis-independent, preserves $\mathcal{G}$-invariance, and coincides with $k$ when $k$ is already PSD.

Let $\mathcal{S} = \{\boldsymbol{x}_1, \dots, \boldsymbol{x}_N\}$ be finite, and let $\mathcal{G}$ act on $\mathcal{S}$. Consider any symmetric kernel $k$ on $\mathcal{S}$ with Gram matrix $\boldsymbol{K} \in \mathbb{R}^{N \times N}$ (possibly indefinite) given by $\boldsymbol{K}_{ij} = k(\boldsymbol{x}_i, \boldsymbol{x}_j)$. We define $k_+$ as the kernel corresponding to the Frobenius-nearest PSD projection of $\boldsymbol{K}$ (Higham, 1988).

---

[7]Only step 2 in Section 2.1 may fail if $\boldsymbol{K}_t + \sigma_0^2 \boldsymbol{I}_t$ is non-invertible. One can use a pseudo-inverse or a very large $\sigma_0$, but the latter makes the posterior variance nearly flat, degenerating the procedure into blind exploitation.

**Lemma 7** (Frobenius PSD projection and explicit form (Higham, 1988)). *The optimization problem* $\boldsymbol{K}_+ := \arg\min_{\boldsymbol{P} \succeq 0} \|\boldsymbol{P} - \boldsymbol{K}\|_F$ *has a unique solution and, for any eigendecomposition* $\boldsymbol{K} = \boldsymbol{Q}\boldsymbol{\Lambda}\boldsymbol{Q}^\top$*, it is given by*

$$\boldsymbol{K}_+ = \boldsymbol{Q}\max(0,\boldsymbol{\Lambda})\,\boldsymbol{Q}^\top,$$

*where* $\max(0,\cdot)$ *acts entrywise on* $\boldsymbol{\Lambda}$*. In particular, the matrix* $\boldsymbol{K}_+$ *depends only on* $\boldsymbol{K}$ *(not on the chosen eigenbasis), satisfies* $\boldsymbol{K}_+ \succeq 0$*, and* $\boldsymbol{K}_+ = \boldsymbol{K}$ *iff* $\boldsymbol{K} \succeq 0$*.*

We *define* $k_+$, the (Frobenius) PSD projection of $k$, as:

$$k_+(x_i, x_j) := (\boldsymbol{K}_+)_{ij}, \qquad i, j \in [N]. \tag{11}$$

**Inheritance of $\mathcal{G}$-invariance.** Each element $g \in \mathcal{G}$ induces a permutation of the elements of $\mathcal{S}$: let $\pi_g$ be the permutations of the integers $j \in \{1, \ldots, N\}$ defined by $g\boldsymbol{x}_j = \boldsymbol{x}_{\pi_g(j)}$. Denote by $\boldsymbol{P}_g$ the permutation matrix associated with $\pi_g$. For every vector $\boldsymbol{v}$, the matrix $\boldsymbol{P}_g$ acts as $(\boldsymbol{P}_g\boldsymbol{v})_i = \boldsymbol{v}_{\pi_g^{-1}(i)}$ which is equivalent to the action on canonical vectors $\boldsymbol{P}_g\boldsymbol{e}_j = \boldsymbol{e}_{\pi_g(j)}$ or $(\boldsymbol{P}_g)_{ij} = 1_{i = \pi_g(j)}$.

Invariance in the first component guarantees $k_{\max}(\boldsymbol{x}_{\pi_g(i)}, \boldsymbol{x}_j) = k_{\max}(g\boldsymbol{x}_i, \boldsymbol{x}_j) = k_{\max}(\boldsymbol{x}_i, \boldsymbol{x}_j)$ for every $i, j \in \{1, \ldots, N\}$, i.e., the rows of $\boldsymbol{K} = (k(\boldsymbol{x}_i, \boldsymbol{x}_j))_{i,j}$ are invariant under the permutation $\pi_g$, hence $\boldsymbol{P}_g\boldsymbol{K} = \boldsymbol{K}$. Thus, for any positive integer $m$, $\boldsymbol{P}_g\boldsymbol{K}^m = (\boldsymbol{P}_g\boldsymbol{K})\boldsymbol{K}^{m-1} = \boldsymbol{K}^m$ so for any polynomial $p$ such that $p(0) = 0$, $\boldsymbol{P}_g p(\boldsymbol{K}) = p(\boldsymbol{K})$. Now consider a sequence $(p_n)_n$ of polynomials such that[8] $p_n(0) = 0$ and $|p_n(\lambda) - \max(0, \lambda)| \xrightarrow[n\to\infty]{} 0$ for any $\lambda$ in the spectrum of $\boldsymbol{K}$. In the limit $\boldsymbol{P}_g\boldsymbol{K}_+ = \boldsymbol{K}_+$, hence $k_+$ is invariant under the action of $\mathcal{G}$ on the first variable ($k_+(g\boldsymbol{x}, \boldsymbol{x}') = k_+(\boldsymbol{x}, \boldsymbol{x}')$), and invariance along the second one follows by symmetry ($\boldsymbol{K}_+\boldsymbol{P}_g^\top = \boldsymbol{K}_+$). This shows that $k_+$ inherits from the $\mathcal{G}$-invariance of $k$ (equivalently, $\boldsymbol{P}_g\boldsymbol{K} = \boldsymbol{K} = \boldsymbol{K}\boldsymbol{P}_g^\top$ for all $g$). We collect this result in the next lemma.

**Lemma 8** (Invariance is preserved by the projection). *Consider* $g \in \mathcal{G}$*. If* $P_g\boldsymbol{K} = \boldsymbol{K}$*, then* $P_g\boldsymbol{K}_+ = \boldsymbol{K}_+ = \boldsymbol{K}_+ P_g^\top$*. Hence the projected kernel* $k_+$ *is* $\mathcal{G}$*-invariant on* $\mathcal{S} \times \mathcal{S}$*.*

**Relation to the practical Nyström kernel.** If the set $\mathcal{D} = \{\boldsymbol{x}_1, \ldots, \boldsymbol{x}_n\}$ used to build $k_+^{(\mathcal{D})}$ (Equation (7)) equals the whole domain $\mathcal{D} = \mathcal{S}$, then $k_+^{(\mathcal{D})} = k_+$. Indeed, $k_+^{(\mathcal{D})}(\boldsymbol{x}_i, \boldsymbol{x}_j) = \boldsymbol{K}_{i:}\boldsymbol{K}_+^\dagger \boldsymbol{K}_{:j} = (\boldsymbol{K}\boldsymbol{K}_+^\dagger\boldsymbol{K})_{ij} = (\boldsymbol{K}_+)_{ij}$ on $\mathcal{D} \times \mathcal{D}$, and the latter is the definition of $k_+$ on finite domains.

We now generalize the matrix considerations above using integral operators. The finite-domain construction is recovered as a special case.

### C.3 GENERAL DEFINITION (VIA INTEGRAL OPERATORS THEORY)

We lift the finite-domain construction of the previous subsection to general domains by viewing $k$ as a Hilbert–Schmidt operator and defining $k_+$ as the positive part of $T_k$; this yields a PSD, data-independent kernel that inherits any $\mathcal{G}$-invariance and equals $k$ whenever $k$ is PSD.

Let $(\mathcal{S}, \mathcal{T}, \mu)$ be a probability space. For a measurable, symmetric kernel $k : \mathcal{S} \times \mathcal{S} \to \mathbb{R}$ with $k \in L^2(\mu \otimes \mu)$, let the (compact, self-adjoint) Hilbert-Schmidt operator $T_k : L^2(\mu) \to L^2(\mu)$ be

$$(T_k f)(\boldsymbol{x}) = \int_{\mathcal{S}} k(\boldsymbol{x}, \boldsymbol{x}') f(\boldsymbol{x}') \, d\mu(\boldsymbol{x}').$$

(Note that in the finite-domain case, $f$ is a vector indexed by the domain and if $\mu$ is the uniform measure then $T_k$ is simply multiplication by the Gram matrix $\boldsymbol{K}$ normalized by the domain size.) By the spectral theorem, there exist $(\lambda_i, \phi_i)_{i \geq 1}$ with $\{\phi_i\}$ orthonormal in $L^2(\mu)$ and $(\lambda_i) \in \ell^2$ (possibly of mixed signs) such that $T_k = \sum_{i \geq 1} \lambda_i\, \phi_i \otimes \phi_i$ in $L^2(\mu)$ where for every $u, v \in L^2(\mu)$, $u \otimes v$ is the rank-one operator $L^2(\mu) \to L^2(\mu)$ such that $(u \otimes v)f := \langle f, v \rangle u$ for every $f \in L^2(\mu)$.

---

[8]We can impose $p_n(0) = 0$ since $f(0) = 0$. Indeed, take $p_n(\lambda) = q_n(\lambda) - q_n(0)$ where $q_n$ is a sequence given by Weierstrass' theorem, which converges to $f(\lambda) = \max(0, \lambda)$ on the spectrum of $\boldsymbol{K}$. We have $|p_n(\lambda) - f(\lambda)| \leq |q_n(\lambda) - f(\lambda)| + |q_n(0)|$ and because $f(0) = 0$ we get $|q_n(0)| = |q_n(0) - f(0)| \to 0$.

**Generic definition of $k_+$ via operator theory.** Define the positive part of $T_k = \sum_i \lambda_i \, \phi_i \otimes \phi_i$ by $T_k^+ := \sum_i (\lambda_i)_+ \, \phi_i \otimes \phi_i$, where $(t)_+ = \max\{t, 0\}$. Since $\sum_i ((\lambda_i)_+)^2 \leq \sum_i \lambda_i^2 < \infty$, the series

$$k_+(\boldsymbol{x}, \boldsymbol{x}') := \sum_{i \geq 1} (\lambda_i)_+ \, \phi_i(\boldsymbol{x}) \, \phi_i(\boldsymbol{x}') \quad (\mu \otimes \mu\text{-a.e.}). \tag{12}$$

converges in $L^2(\mu \otimes \mu)$ and defines a kernel $\mu \otimes \mu$-almost everywhere. By construction[9] $T_{k_+} = T_k^+$, hence $k_+$ is PSD as a kernel a.e., and PSD in the operator sense: $\langle f, T_{k_+} f \rangle \geq 0$ for all $f \in L^2(\mu)$. In particular, if $k$ was already PSD (all $\lambda_i \geq 0$), then $k_+ = k$ (up to null sets). It also inherits $\mathcal{G}$-invariance of $k$ if $k$ is indeed invariant (the proof mimics the finite-domain case, we give the full details for completeness in Appendix C.7).

## C.4 FROM THE FINITE-SAMPLE PROJECTION TO THE INTRINSIC LIMIT: WHAT CONVERGES TO WHAT?

We relate the practical, data-dependent Nyström kernel $k_+^{(\mathcal{D})}$ (Equation (7)) to the intrinsic $k_+$: under iid sampling, the empirical spectra of $k_+^{(\mathcal{D})}/|\mathcal{D}|$ converge to that of $T_{k_+}$, with rates under mild moment assumptions. This shows that eigendecay-based regret analysis

**Notations.** Let $X_1, X_2, \cdots \sim \mu$ i.i.d. and $\mathcal{D}_n = \{X_1, \ldots, X_n\}$. We write $\boldsymbol{K}_n := k(\mathcal{D}_n, \mathcal{D}_n)$, $\boldsymbol{K}_n^+ := \arg\min_{\boldsymbol{P} \succeq 0} \|\boldsymbol{P} - \boldsymbol{K}_n\|_F$, $\tilde{\boldsymbol{K}}_n := \boldsymbol{K}_n/n$, and recall that the practical (data-dependent) kernel defined in Equation (7) is

$$k_+^{(\mathcal{D}_n)}(\boldsymbol{x}, \boldsymbol{x}') = k(\boldsymbol{x}, \mathcal{D}_n) \, (\boldsymbol{K}_n^+)^\dagger \, k(\mathcal{D}_n, \boldsymbol{x}').$$

We denote by $\lambda(T)$ the (ordered, nonincreasing, each counted with its multiplicity) sequence of eigenvalues of a compact self-adjoint operator $T$, and by $\delta_2\big(\lambda(T), \lambda(S)\big) := \big(\sum_i |\lambda_i(T) - \lambda_i(S)|^2\big)^{1/2}$ the spectral $\ell_2$ distance. For symmetric matrices $\boldsymbol{M}$, $\lambda(\boldsymbol{M})$ denotes the nonincreasing sequence of eigenvalues of $\boldsymbol{M}$ (with multiplicity) padded with an infinite number of zeros. For a bounded operator $A$, $\|A\|_{\mathrm{HS}}$ and $\|A\|_{\mathrm{op}}$ denote the Hilbert-Schmidt and operator norms, respectively. We include in Appendix C.5 a reminder on the different notions of norms and convergence, and we now recall the essentials.

**Relations between convergence notions.** For compact self-adjoint operators: (i) $\max\big(\delta_2\big(\lambda(T_n), \lambda(T)\big), \|T_n - T\|_{\mathrm{op}}\big) \leq \|T_n - T\|_{\mathrm{HS}}$ (Reed & Simon, 1972; Bhatia & Elsner, 1994); (ii) converse inequalities do not hold in infinite dimension (see Appendix C.5 for examples). Thus, HS convergence is the strongest notion of convergence we manipulate here.

We now present convergence guarantees of the data-dependent construction $k_+^{(\mathcal{D}_n)}/n$ to the intrinsic $k_+$ under progressively stronger assumptions. With minimal assumptions we obtain almost-sure spectral consistency in the $\delta_2$ metric; with stronger assumptions we obtain quantitative rates in HS norm (hence also spectral $\ell_2$ in probability).

**(a) Weak a.s. spectral consistency of positive parts (minimal assumptions).**

**Proposition 9.** *Assume the symmetric (not necessarily PSD) kernel $k$ is in $L^2(\mu \otimes \mu)$ so that $T_k$ is Hilbert-Schmidt. Let $\widehat{S}_n : L^2(\mu_n) \to L^2(\mu_n)$ be the integral operator with kernel $k_+^{(\mathcal{D}_n)}(\boldsymbol{x}, \boldsymbol{x}')/n$ defined by:*

$$(\widehat{S}_n f)(\boldsymbol{x}) = \frac{1}{n} \sum_{j=1}^n k_+^{(\mathcal{D}_n)}(\boldsymbol{x}, X_j) f(X_j). \tag{13}$$

*Assume the $X_i$ are pairwise distinct almost surely. Then, almost surely,*

$$\delta_2\Big(\lambda(\widehat{S}_n), \; \lambda(T_{k_+})\Big) \underset{n \to \infty}{\longrightarrow} 0.$$

---

[9]Indeed, by definition $(T_{k_+} f)(\boldsymbol{x}) = \int_{\mathcal{S}} \Big(\sum_{i \geq 1} (\lambda_i)_+ \phi_i(\boldsymbol{x}) \phi_i(\boldsymbol{x}')\Big) f(\boldsymbol{x}') \, d\mu(\boldsymbol{x}') = \sum_{i \geq 1} (\lambda_i)_+ \langle f, \phi_i \rangle \, \phi_i(\boldsymbol{x}) = \Big(\Big(\sum_{i \geq 1} (\lambda_i)_+ \phi_i \otimes \phi_i\Big) f\Big)(\boldsymbol{x}) = (T_k^+ f)(\boldsymbol{x})$.

*Proof.* Let $\boldsymbol{K}_n$ be the empirical operator on $\mathbb{R}^n$ with matrix $\frac{1}{n}(k(X_i, X_j))_{i,j}$ and let $\lambda(\boldsymbol{K}_n)$ be its ordered spectrum (nonincreasing, with multiplicity) padded with an infinite number of zeros. Theorem 3.1 of Koltchinskii & Giné (2000) shows that $\delta_2(\lambda(\boldsymbol{K}_n), \lambda(T_k)) \to 0$ as $n \to \infty$.

Let $\boldsymbol{K}_n^+$ be the positive part of $\boldsymbol{K}_n$ (i.e., its Frobenius PSD projection). Since $\lambda \mapsto \max(0, \lambda)$ is 1-Lipschitz, we have for any operators $T, S$:

$$\delta_2(\lambda(T_+), \lambda(S_+)) = \sum_i |\max(0, \lambda_i(T)) - \max(0, \lambda_i(S))| \leq \sum_i |\lambda_i(T) - \lambda_i(S)| = \delta_2(\lambda(T), \lambda(S)).$$

We deduce that $\delta_2(\lambda(\boldsymbol{K}_n^+), \lambda(T_{k_+})) \to 0$ as $n \to \infty$.

It remains to observe that the spectrum of $\boldsymbol{K}_n^+$ as an operator on $\mathbb{R}^n$ is the same as $\widehat{S}_n : L^2(\mu_n) \to L^2(\mu_n)$. This identification is standard (e.g., see above Equation 1.2 in Koltchinskii & Giné (2000)). For completeness, we include the formal arguments of Koltchinskii & Giné (2000) in Lemma 12, which shows that we can identify the spectrum of $k_+^{(\mathcal{D}_n)}(\mathcal{D}_n, \mathcal{D}_n)/n$ with the one of $\boldsymbol{K}_n^+$ a.s. if the iid $X_i \sim \mu$ are pairwise distinct a.s, which is true as soon as $\mu$ is non-atomic; otherwise one can index the *distinct* atoms and work in $\mathbb{R}^m$ with $m = \#\mathrm{supp}(\mu_n)$, obtaining the same spectral identity on that subspace. $\square$

**(b) Expected HS convergence with $\mathcal{O}(n^{-1/2})$ rate (stronger assumption).** Define the empirical integral operator $(T_n f)(\boldsymbol{x}) := \frac{1}{n} \sum_{i=1}^n k(\boldsymbol{x}, X_i) f(X_i)$ and $D_n := T_n - T_k$. Let $(\lambda_i, \phi_i)_{i \geq 1}$ be an eigensystem of $T_k$ in $L^2(\mu)$. Assume the following fourth-order summability condition holds:

$$C := \sum_{i,j \geq 1} \lambda_i^2 \int_{\mathcal{S}} \phi_i(\boldsymbol{x})^2 \phi_j(\boldsymbol{x})^2 \, d\mu(\boldsymbol{x}) < \infty. \tag{14}$$

**Proposition 10** (Expected HS rate). *Under $k \in L^2(\mu \otimes \mu)$ and (14),*

$$\mathbb{E}\big[\|D_n\|_{\mathrm{HS}}^2\big] \leq \frac{C}{n}, \qquad \mathbb{E}\big[\|D_n\|_{\mathrm{HS}}\big] \leq \sqrt{\frac{C}{n}}.$$

*Consequently, $\|D_n\|_{\mathrm{HS}} = \mathcal{O}_{\mathbb{P}}(n^{-1/2})$ and therefore using the same notations as in Proposition 9*

$$\delta_2\big(\lambda(\boldsymbol{K}_n^+), \lambda(T_k^+)\big) = \mathcal{O}_{\mathbb{P}}(n^{-1/2}), \qquad \delta_2\big(\lambda(\widehat{S}_n), \lambda(T_{k_+})\big) = \mathcal{O}_{\mathbb{P}}(n^{-1/2}).$$

*Proof.* Fix any $f \in L^2(\mu)$. By Fubini-Tonelli for non-negative functions, we have:

$$\mathbb{E}\big[\|D_n f\|_{L^2(\mu)}^2\big] = \int_{\mathcal{S}} \mathbb{E}\Big[\big((D_n f)(\boldsymbol{x})\big)^2\Big] d\mu(\boldsymbol{x}).$$

By definition

$$(D_n f)(\boldsymbol{x}) = \frac{1}{n} \sum_{i=1}^n k(\boldsymbol{x}, X_i) f(X_i) - \int_{\mathcal{S}} k(\boldsymbol{x}, \boldsymbol{x}') f(\boldsymbol{x}') \, d\mu(\boldsymbol{x}')$$

where the randomness comes from the i.i.d. $X_i \sim \mu$. Hence $\mathbb{E}\big[(D_n f)(\boldsymbol{x})\big] = 0$ and for any fixed $\boldsymbol{x}$

$$\mathbb{E}\Big[\big((D_n f)(\boldsymbol{x})\big)^2\Big] = \mathrm{Var}\big((D_n f)(\boldsymbol{x})\big) = \frac{1}{n} \mathrm{Var}\big(k(\boldsymbol{x}, X) f(X)\big) \leq \frac{1}{n} \int_{\mathcal{S}} k(\boldsymbol{x}, \boldsymbol{x}')^2 f(\boldsymbol{x}')^2 \, d\mu(\boldsymbol{x}').$$

The Hilbert-Schmidt spectral theorem gives the expansion $k(\boldsymbol{x}, \boldsymbol{x}') = \sum_i \lambda_i \phi_i(\boldsymbol{x}) \phi_i(\boldsymbol{x}')$ in $L^2(\mu \otimes \mu)$, with $(\lambda_i)_i \in \ell^2$ and $(\phi_i)_i$ an orthonormal set of $L^2(\mu)$ (see Equation 3.2 in Koltchinskii & Giné (2000), Corollary 5.4 in Conway (2007)). Thus

$$\int_{\mathcal{S}} \mathbb{E}\Big[\big((D_n f)(\boldsymbol{x})\big)^2\Big] d\mu(\boldsymbol{x}) \leq \frac{1}{n} \int_{\mathcal{S}} k(\boldsymbol{x}, \boldsymbol{x}')^2 f(\boldsymbol{x}')^2 \, d\mu(\boldsymbol{x}') d\mu(\boldsymbol{x})$$

$$= \sum_{i,j} \lambda_i \lambda_j \int_{\mathcal{S}} \phi_i(\boldsymbol{x}') \phi_j(\boldsymbol{x}') f(\boldsymbol{x}')^2 \underbrace{\langle \phi_i, \phi_j \rangle}_{=1_{i=j}} d\mu(\boldsymbol{x}')$$

$$= \sum_i \lambda_i^2 \int_{\mathcal{S}} \phi_i(\boldsymbol{x}')^2 f(\boldsymbol{x}')^2 d\mu(\boldsymbol{x}').$$

Taking $f = \phi_j$ for a fixed $j$ yields

$$\mathbb{E}\big[\|D_n \phi_j\|^2_{L^2(\mu)}\big] \leq \frac{1}{n} \sum_i \lambda_i^2 \int_{\mathcal{S}} \phi_i(\boldsymbol{x}')^2 \phi_j(\boldsymbol{x}')^2 d\mu(\boldsymbol{x}').$$

Since $\|D_n f\|^2_{\mathrm{HS}} = \sum_j \|D_n \phi_j\|^2_{L^2(\mu)}$, we get the main claim:

$$\mathbb{E}\big[\|D_n\|^2_{\mathrm{HS}}\big] \ \leq \ \frac{C}{n}.$$

Jensen gives the bound for $\mathbb{E}\|D_n\|_{\mathrm{HS}}$. Finally, $\delta_2(\lambda(\boldsymbol{K}_n), \lambda(T_k)) \leq \|D_n\|_{\mathrm{HS}}$ (Hoffman-Wielandt inequality in infinite dimension (Bhatia & Elsner, 1994)), and $\lambda \mapsto \max(0, \lambda)$ is 1-Lipschitz on $\mathbb{R}$, hence the spectral bound probability claim using Markov's inequality, and Lemma 12 transfers this claims to $\widehat{S}_n$. □

**Remark 11** (On assumption (14)). *Condition* (14) *is a fourth-order integrability requirement that controls eigenfunction overlaps. It is standard in random Nyström analyses (see, e.g., Equations (4.3) and (4.11) of Koltchinskii & Giné (2000)) and stronger than $k \in L^2$, but it yields a dimension-free $\mathcal{O}(n^{-1/2})$ rate in HS norm.*

**(c) High-probability HS rates (heavier but more precise).** Under slightly stronger $L^4$-type conditions on eigenfunctions, the section 4 in Koltchinskii & Giné (2000) gives more more precise statements on the rates in Proposition 10, and we directly refer the reader to it.

**Application to $k_{\max}$ and to the BO kernels in the paper.** When $k = k_{\max}$ is bounded on a compact domain $\mathcal{S}$ (as in all our experiments), $k \in L^2(\mu \otimes \mu)$ for any probability measure $\mu$ on $\mathcal{S}$, so $T_{k_{\max}}$ is Hilbert-Schmidt and Proposition 9 applies. In particular, the integral operator associated with $k_+^{(\mathcal{D}_n)}/n$, called $\widehat{S}_n$ (Equation (13)) satisfies

$$\delta_2\Big(\lambda(\widehat{S}_n), \ \lambda(T_{k_+})\Big) \ \xrightarrow[n \to \infty]{\text{a.s.}} \ 0.$$

This clarifies the two objects introduced in the main text: the *intrinsic* $k_+$ is the unique data-independent target, while the *practical* kernel $k_+^{(\mathcal{D}_n)}$ (finite PSD projection + Nyström) is an on-path approximation whose spectrum converges (once normalized by $n$) to that of $k_+$ under i.i.d. sampling.

The following subsections are only optional complementary materials added to help building intuitions on the convergence results stated above.

### C.5 REMINDERS ON THE DIFFERENT TYPE OF CONVERGENCES FOR BOUNDED LINEAR OPERATORS

This subsection recalls standard notions of operator convergence, included only as background to help build intuition for the convergence results above.

**Definitions (operator norm, HS norm, spectral distance).** Let $\mathcal{H}$ be a separable Hilbert space with orthonormal basis $\{e_i\}_{i \geq 1}$. For a bounded linear operator $T : \mathcal{H} \to \mathcal{H}$,

$$\|T\|_{\mathrm{op}} := \sup_{\|f\|_{\mathcal{H}}=1} \|Tf\|_{\mathcal{H}}, \qquad \|T\|_{\mathrm{HS}} := \Big(\sum_{i \geq 1} \|Te_i\|^2_{\mathcal{H}}\Big)^{1/2}.$$

The HS norm is basis-independent. When $T$ is an *integral* operator with kernel $k \in L^2(\mu \otimes \mu)$ on $L^2(\mu)$ (Reed & Simon, 1972)

$$\|T\|^2_{\mathrm{HS}} = \iint_{\mathcal{S} \times \mathcal{S}} |k(x,y)|^2 \, d\mu(x) \, d\mu(y).$$

For finite matrices, $\|A\|_{\mathrm{HS}} = \|A\|_F$ (Frobenius). We say $T_n \to T$ in HS norm if $\|T_n - T\|_{\mathrm{HS}} \to 0$, and we say $T_n \to T$ spectrally if $\delta_2(\lambda(T_n), \lambda(T)) \to 0$, where we recall that $\lambda(T)$ is the *ordered* eigenvalues of a compact self-adjoint operator $T$, and where the spectral $\ell_2$-distance is $\delta_2(\lambda(T), \lambda(S)) := \big(\sum_i |\lambda_i(T) - \lambda_i(S)|^2\big)^{1/2}$.

**Which convergences matter, and how they relate (reminders on well-known facts).** We compare three notions: (i) *operator norm* convergence $\|T_n - T\|_{\mathrm{op}} \to 0$; (ii) *Hilbert-Schmidt (HS)* convergence $\|T_n - T\|_{\mathrm{HS}} \to 0$; (iii) *spectral* convergence in $\delta_2$, i.e., $\delta_2\big(\lambda(T_n), \lambda(T)\big) := \big(\sum_i |\lambda_i(T_n) - \lambda_i(T)|^2\big)^{1/2} \to 0$, where $\lambda(\cdot)$ denotes the ordered eigenvalues of a compact self-adjoint operator. We recall the following well-known facts, useful to grasp the convergence results we state next.

**(1) HS $\Longrightarrow$ spectral $\delta_2$.** For compact self-adjoint operators the (infinite-dimensional) Hoffman-Wielandt inequality yields (Bhatia & Elsner, 1994)

$$\delta_2\big(\lambda(T_n), \lambda(T)\big) \leq \|T_n - T\|_{\mathrm{HS}}.$$

**(2) HS $\Longrightarrow$ operator norm.** For every Hilbert-Schmidt operator $S$, $\|S\|_{\mathrm{op}} \leq \|S\|_{\mathrm{HS}}$. Indeed for unit vectors $x, y \in H$, using $x = \sum_i \langle x, e_i \rangle e_i$, we have $\langle Sx, y \rangle = \sum_{i \in I} \langle x, e_i \rangle \langle Se_i, y \rangle$. By Cauchy-Schwarz:

$$|\langle Sx, y \rangle| \leq \Big( \sum_{i \in I} |\langle x, e_i \rangle|^2 \Big)^{1/2} \Big( \sum_{i \in I} |\langle Se_i, y \rangle|^2 \Big)^{1/2}.$$

The first factor equals $\|x\| = 1$, and for the second we use $|\langle Se_i, y \rangle| \leq \|Se_i\| \|y\| = \|Se_i\|$ to get

$$\sum_{i \in I} |\langle Se_i, y \rangle|^2 \leq \sum_{i \in I} \|Se_i\|^2 = \|S\|_{\mathrm{HS}}^2.$$

Hence $|\langle Sx, y \rangle| \leq \|S\|_{\mathrm{HS}}$. Taking the supremum over all unit $y$ gives

$$\|Sx\| = \sup_{\|y\|=1} |\langle Sx, y \rangle| \leq \|S\|_{\mathrm{HS}},$$

and then taking the supremum over all unit $x$ yields

$$\|S\|_{\mathrm{op}} = \sup_{\|x\|=1} \|Sx\| \leq \|S\|_{\mathrm{HS}}.$$

**(3) Spectral $\delta_2$ does *not* imply HS nor operator norm.** Even if eigenvalues match in $\ell_2$, the operators may be far in norm because eigenvectors can rotate. Let $T = \mathrm{diag}(1, 1/2, 1/3, \ldots)$ in the canonical basis $(e_i)_{i \geq 1}$, and let $U_n$ swap $e_1$ and $e_n$. Set $T_n := U_n T U_n^*$. Then $\lambda(T_n) = \lambda(T)$ for all $n$ (same ordered spectrum), so $\delta_2(\lambda(T_n), \lambda(T)) = 0$. Yet $\|(T_n - T)e_1\| = \|(U_n T U_n^* - T)e_1\| = \|(1/n - 1)e_1\| = 1 - 1/n$, hence $\|T_n - T\|_{\mathrm{op}} \geq 1 - 1/n \to 1$ and, a fortiori, $\|T_n - T\|_{\mathrm{HS}} \not\to 0$.

**(4) Operator norm does *not* imply spectral $\delta_2$.** Let $T = 0$ and $T_n$ be diagonal with the first $m_n$ entries equal to $\varepsilon_n$ and the rest 0. Choose $\varepsilon_n := n^{-1/2}$ and $m_n := n$. Then $\|T_n\|_{\mathrm{op}} = \varepsilon_n \to 0$ but $\delta_2\big(\lambda(T_n), \lambda(T)\big) = \big(\sum_{i=1}^{m_n} \varepsilon_n^2\big)^{1/2} = \sqrt{n \cdot (1/n)} = 1$.

**(5) Two useful corollaries.** (a) Spectral $\delta_2$-convergence implies convergence of the *largest* eigenvalue, since $\sup_i |\lambda_i(T_n) - \lambda_i(T)| \leq \delta_2(\lambda(T_n), \lambda(T))$. (b) Operator-norm convergence forces uniform eigenvalue deviations to vanish by Weyl's inequality: $\sup_i |\lambda_i(T_n) - \lambda_i(T)| \leq \|T_n - T\|_{\mathrm{op}}$, but it does *not* control the $\ell_2$-sum of all deviations.

*Takeaway.* HS is the strongest notion here: it simultaneously implies spectral $\delta_2$-convergence (and thus convergence of eigenvalue-based quantities) and operator-norm convergence. The converses fail in infinite dimension because eigenvectors can drift and an infinite number of tiny eigenvalue errors can accumulate.

## C.6 IDENTIFICATION OF THE SPECTRUM OF AN EMPIRICAL OPERATOR IN $L^2(\mu_n)$ AND ITS MATRIX COUNTERPART

Here we show how the spectrum of the empirical operator can be identified with that of its matrix form. This is complementary material meant to clarify how operator-level and matrix-level viewpoints connect (which is useful, e.g., in the proof of Proposition 9).

**Lemma 12** (Empirical Nyström spectral identity). *Let $\boldsymbol{K}_n := \frac{1}{n}\big(k(\boldsymbol{x}_i, \boldsymbol{x}_j)\big)_{i,j=1}^n$ and let $\boldsymbol{K}_n^+$ be its spectral positive part (the Frobenius-nearest PSD projection). Define the empirical measure $\mu_n := \frac{1}{n}\sum_{i=1}^n \delta_{\boldsymbol{x}_i}$ and the Nyström kernel*

$$k_+^{(\mathcal{D}_n)}(\boldsymbol{x}, \boldsymbol{x}') \;=\; k(\boldsymbol{x}, \mathcal{D}_n)\,(\boldsymbol{K}_n^+)^\dagger\, k(\mathcal{D}_n, \boldsymbol{x}').$$

*Let $\widehat{S}_n : L^2(\mu_n) \to L^2(\mu_n)$ be the integral operator with kernel $k_+^{(\mathcal{D}_n)}(\boldsymbol{x}, \boldsymbol{x}')/n$, i.e.*

$$(\widehat{S}_n f)(\boldsymbol{x}) \;=\; \frac{1}{n}\sum_{j=1}^n k_+^{(\mathcal{D}_n)}(\boldsymbol{x}, \boldsymbol{x}_j)\, f(\boldsymbol{x}_j).$$

*The map $E : L^2(\mu_n) \to \mathbb{R}^n$, $Ef := \frac{1}{\sqrt{n}}\big(f(\boldsymbol{x}_1), \ldots, f(\boldsymbol{x}_n)\big)^\top$, is an isometry: $\|Ef\|_{\mathbb{R}^n} = \|f\|_{L^2(\mu_n)}$, and we have the intertwining identity*

$$E\,\widehat{S}_n \;=\; \boldsymbol{K}_n^+\, E.$$

*If, in addition, the sample points $\boldsymbol{x}_1, \ldots, \boldsymbol{x}_n$ are pairwise distinct, then $E$ is an isometric isomorphism (hence invertible) and*

$$\lambda\big(\widehat{S}_n\big) \;=\; \lambda\big(\boldsymbol{K}_n^+\big) \;=\; \lambda\big(k_+^{(\mathcal{D}_n)}(\mathcal{D}_n, \mathcal{D}_n)/n\big).$$

*Proof.* First note the on-sample identity $k_+^{(\mathcal{D}_n)}(\boldsymbol{x}_i, \boldsymbol{x}_j) \;=\; (\boldsymbol{K}^+)_{ij}$ for the unscaled $\boldsymbol{K} = (k(\boldsymbol{x}_i, \boldsymbol{x}_j))_{i,j}$, which follows from $\boldsymbol{K}(\boldsymbol{K}^+)^\dagger \boldsymbol{K} = \boldsymbol{K}^+$. Hence $k_+^{(\mathcal{D}_n)}(\mathcal{D}_n, \mathcal{D}_n) = \boldsymbol{K}^+$ and therefore $k_+^{(\mathcal{D}_n)}(\mathcal{D}_n, \mathcal{D}_n)/n = \boldsymbol{K}_n^+$.

For $f \in L^2(\mu_n)$ and each $i \in \{1, \ldots, n\}$,

$$\sqrt{n}\,\big(E\widehat{S}_n f\big)_i = (\widehat{S}_n f)(\boldsymbol{x}_i) = \frac{1}{n}\sum_{j=1}^n k_+^{(\mathcal{D}_n)}(\boldsymbol{x}_i, \boldsymbol{x}_j)\, f(\boldsymbol{x}_j) = \sum_{j=1}^n (\boldsymbol{K}_n^+)_{ij}\, f(X_j) = \sqrt{n}\,\big(\boldsymbol{K}_n^+ Ef\big)_i,$$

which proves $E\,\widehat{S}_n = \boldsymbol{K}_n^+ E$. Since $E$ is an isometry by definition of the $L^2(\mu_n)$ inner product, if the $X_i$ are pairwise distinct then $E$ is bijective and conjugates $\widehat{S}_n$ with $\boldsymbol{K}_n^+$, so the spectra (with multiplicities) coincide. $\qquad\square$

## C.7 PROOF OF $\mathcal{G}$-INVARIANCE OF $k_+$ FOR GENERAL DOMAINS

We conclude this appendix with the formal proof that $k_+$ defined in (12) inherits from any group-invariance of $k$. This proof is not needed for the main results but is included for completeness. It makes explicit why $k_+$ preserves any $\mathcal{G}$-invariance of $k$. The proof follows the one for finite domains but is heavier in notations because it is now stated using integral operators to generalize the matrix manipulations of finite domains. For finite domains, denoting by $\boldsymbol{K}$ the Gram matrix of $k$ over the whole domain and $\boldsymbol{P}_g$ the permutation matrix induced by the action of $g \in \mathcal{G}$ on the domain, invariance of $k$ is equivalent to $\boldsymbol{P}_g \boldsymbol{K} = \boldsymbol{K} \boldsymbol{P}_g^\top = \boldsymbol{K}$. Thus any polynomial $p(\boldsymbol{K})$ of $\boldsymbol{K}$ such that $p(0) = 0$ inherits from this invariance since we still have $\boldsymbol{P}_g p(\boldsymbol{K}) = p(\boldsymbol{K})\boldsymbol{P}_g^\top = p(\boldsymbol{K})$. And at the limit, we get invariance of $\boldsymbol{K}_+$. Here, we mimic this proof, and we start by introducing the equivalent integral operator form of the characterization $\boldsymbol{P}_g \boldsymbol{K} = \boldsymbol{K}\boldsymbol{P}_g^\top = \boldsymbol{K}$ for general domains.

**Lemma 13** (Kernel invariance $\Longleftrightarrow$ operator commutation). *Let $(\mathcal{S}, \mathcal{T}, \mu)$ be a probability space and let $\mathcal{G}$ act measurably on $\mathcal{S}$. Assume $\mu$ is $\mathcal{G}$-invariant. Let $U_g : L^2(\mu) \to L^2(\mu)$ be the unitary representation $(U_g f)(\boldsymbol{x}) := f(g^{-1}\boldsymbol{x})$. Let $k \in L^2(\mu \otimes \mu)$ be a symmetric kernel with integral operator $(T_k f)(\boldsymbol{x}) = \int_{\mathcal{S}} k(\boldsymbol{x}, \boldsymbol{x}') f(\boldsymbol{x}')\, d\mu(\boldsymbol{x}')$. Then the following are equivalent:*

(i) *$k$ is argumentwise $\mathcal{G}$-invariant: $k(g\boldsymbol{x}, \boldsymbol{x}') = k(\boldsymbol{x}, g\boldsymbol{x}') = k(\boldsymbol{x}, \boldsymbol{x}')$ for $\mu \otimes \mu$-a.e. $(\boldsymbol{x}, \boldsymbol{x}')$ and all $g \in \mathcal{G}$.*
(ii) *$T_k$ satisfies $U_g T_k = T_k U_g = T_k$ on $L^2(\mu)$ for all $g \in \mathcal{G}$.*

*Proof. (i)$\Rightarrow$(ii).* For any $f \in L^2(\mu)$,

$$(U_g T_k f)(\boldsymbol{x}) = (T_k f)(g^{-1}\boldsymbol{x}) = \int k(g^{-1}\boldsymbol{x}, \boldsymbol{x}')\, f(\boldsymbol{x}')\, d\mu(\boldsymbol{x}').$$

By invariance of $k$ in the first argument $U_g T_k = T_k$. Hence $T_k^* U_g^* = T_k^*$ and $T_k^* = T_k$ (self-adjoint) and $U_g^* = U_{g^{-1}}$ so $T_k U_{g^{-1}} = T_k$. This is true for all $g \in \mathcal{G}$ hence $U_g T_k = T_k U_g = T_k$.

*(ii)⇒(i).* For $\varphi, \psi \in L^2(\mu)$,

$$\iint k(\boldsymbol{x}, \boldsymbol{x}') \, \varphi(\boldsymbol{x}) \psi(\boldsymbol{x}') \, d\mu(\boldsymbol{x}) d\mu(\boldsymbol{x}') = \langle \varphi, T_k \psi \rangle = \langle \varphi, T_k U_g \psi \rangle.$$

Expanding the last inner product, we get by change of variable and invariance of $\mu$

$$\iint k(\boldsymbol{x}, \boldsymbol{x}') \, \varphi(\boldsymbol{x}) \psi(g^{-1}\boldsymbol{x}') \, d\mu(\boldsymbol{x}) d\mu(\boldsymbol{x}') = \iint k(\boldsymbol{x}, g\boldsymbol{x}') \, \varphi(\boldsymbol{x}) \psi(\boldsymbol{x}') \, d\mu(\boldsymbol{x}) d\mu(\boldsymbol{x}').$$

Hence for all $\varphi, \psi$, $\iint [k(\boldsymbol{x}, \boldsymbol{x}') - k(\boldsymbol{x}, g\boldsymbol{x}')] \, \varphi(\boldsymbol{x}) \psi(\boldsymbol{x}') \, d\mu(\boldsymbol{x}) d\mu(\boldsymbol{x}') = 0$, which implies $k(\boldsymbol{x}, g\boldsymbol{x}') = k(\boldsymbol{x}, \boldsymbol{x}') \, \mu \otimes \mu$-a.e. Symmetry implies argumentwise $\mathcal{G}$-invariance. $\qquad\square$

We now show that $U_g T = T$ is preserved if we apply a function $f$ such that $f(0) = 0$ to the spectrum of $T$.

**Lemma 14** (Borel functional calculus preserves invariance)**.** *Let $T$ be a self-adjoint compact operator on a Hilbert space $\mathcal{H}$ with eigendecomposition $T = \sum_i \lambda_i \phi_i \otimes \phi_i$, and let $\{U_g\}_{g \in \mathcal{G}}$ be a unitary representation such that $U_g T = T U_g = T$ for all $g \in \mathcal{G}$. For a bounded Borel function $f : \mathbb{R} \to \mathbb{R}$, define $f(T) = \sum_i f(\lambda_i) \phi_i \otimes \phi_i$. Then for such $f$ with $f(0) = 0$, we have*

$$U_g f(T) = f(T) U_g = f(T) \qquad \text{for all } g \in \mathcal{G}.$$

*Proof.* **Proof sketch:** The assumption $U_g T = T$ forces $U_g$ to act as the identity on each nonzero eigenspace of $T$, which directly yields $U_g f(T) = f(T)$ for any bounded Borel $f$ with $f(0) = 0$.

**Step 1 (spectral decomposition for compact self-adjoint $T$ without measures).** Since $T$ is compact and self-adjoint, its spectrum is $\sigma(T) = \{0\} \cup \{\lambda_n : n \in I\}$ where $I$ is finite or countable, each $\lambda_n \neq 0$ is an eigenvalue of finite multiplicity, and $\lambda_n \to 0$ if infinite. Let $E_\lambda$ denote the eigenspace for $\lambda \neq 0$, and let $E_0 = \ker T$. We have the orthogonal decomposition

$$\mathcal{H} = E_0 \oplus \bigoplus_{\lambda \in \sigma(T) \setminus \{0\}} E_\lambda,$$

and $T$ acts as scalar multiplication on each $E_\lambda$: $T|_{E_\lambda} = \lambda \operatorname{Id}_{E_\lambda}$, $T|_{E_0} = 0$. Let $P_\lambda$ be the orthogonal projector onto $E_\lambda$ (for $\lambda \neq 0$) and $P_0$ onto $E_0$. Then for every $v \in \mathcal{H}$ with expansion $v = v_0 + \sum_{\lambda \neq 0} v_\lambda$ ($v_\lambda := P_\lambda v$), we have

$$Tv = \sum_{\lambda \neq 0} \lambda \, v_\lambda.$$

**Step 2 ($U_g$ fixes each nonzero eigenspace pointwise).** From $U_g T = T$ we get, for any $v \in E_\lambda$ with $\lambda \neq 0$,

$$\lambda U_g v = U_g(Tv) = Tv = \lambda v,$$

hence $U_g v = v$. Thus $U_g$ acts as the identity on each $E_\lambda$ ($\lambda \neq 0$). Equivalently, $U_g P_\lambda = P_\lambda U_g = P_\lambda$ for all $\lambda \neq 0$. (There is no restriction on $U_g$ inside $E_0 = \ker T$.)

**Step 3 (defining $f(T)$ for bounded Borel $f$ with $f(0) = 0$).** Because $\sigma(T) \setminus \{0\}$ is at most countable and $T$ is diagonal on $\{E_\lambda\}$, we can define $f(T)$ by applying $f$ on the spectrum of $T$ as

$$f(T) \, v := \sum_{\lambda \in \sigma(T) \setminus \{0\}} f(\lambda) \, v_\lambda, \qquad v = v_0 + \sum_{\lambda \neq 0} v_\lambda, \; v_\lambda \in E_\lambda.$$

The series converges in norm since the $E_\lambda$ are mutually orthogonal and $\|f(T)v\|^2 = \sum_{\lambda \neq 0} |f(\lambda)|^2 \|v_\lambda\|^2 \leq \left( \sup_{\lambda \neq 0} |f(\lambda)|^2 \right) \sum_{\lambda \neq 0} \|v_\lambda\|^2 \leq \|f\|_\infty^2 \|v\|^2$. Thus $f(T)$ is a bounded operator with $\|f(T)\| \leq \|f\|_\infty$. (When $f(0) = 0$, there is no contribution on $E_0$.)

**Step 4 (invariance and commutation).** For $v = v_0 + \sum_{\lambda \neq 0} v_\lambda$ as above and any $g \in \mathcal{G}$, Step 2 gives $U_g v = U_g v_0 + \sum_{\lambda \neq 0} v_\lambda$ and $P_\lambda U_g = P_\lambda$ for $\lambda \neq 0$. Hence

$$U_g f(T) \, v = U_g \Big( \sum_{\lambda \neq 0} f(\lambda) \, v_\lambda \Big) = \sum_{\lambda \neq 0} f(\lambda) \, U_g v_\lambda = \sum_{\lambda \neq 0} f(\lambda) \, v_\lambda = f(T) \, v,$$

i.e., $U_g f(T) = f(T)$. In particular $U_g f(T) = f(T) U_g = f(T)$ for all $g \in \mathcal{G}$. $\qquad\square$

**Consequence.** If $k$ is $\mathcal{G}$-invariant, then so is $k_+$ (Equation (12)).

## D EIGENDECAY COMPARISON

In this appendix, we discuss in more details the empirical observations made in Section 5 and formally derive some inequalities between Schatten norms of integral operators associated with $k_{\text{avg}}$ and $k_+$.

### D.1 EMPIRICAL OBSERVATIONS

Here, we further discuss the empirical spectra reported in Figure 4 (middle and right columns).

**Computation of spectra.** The normalized Gram matrices $\boldsymbol{K}/n$ (where $\boldsymbol{K} = (k(\boldsymbol{x}_i, \boldsymbol{x}_j))_{1 \leq i,j \leq n}$) reported in Figure 4 are computed from $n = 3000$ i.i.d. samples $\boldsymbol{x}_i \in \mathcal{S}$. We compare the spectra obtained with $k \in \{k_{\text{b}}, k_{\text{avg}}, k_+^{(\mathcal{D})}\}$ with $\mathcal{D} = \{\boldsymbol{x}_1, \ldots, \boldsymbol{x}_n\}$ and each $\boldsymbol{x}_i$ being chosen uniformly in $\mathcal{S} = [-1, 1]$. We also report the spectrum of $k_{\text{b}}$ when observations $\boldsymbol{x}_i$ are instead sampled from an alternative domain $\mathcal{S}'$ of reduced volume, chosen such that $\text{vol}(\mathcal{S}') = \text{vol}(\mathcal{S})/|\mathcal{G}|$. Finally, note that because $\mathcal{D}$ is a set of i.i.d. observations, the spectrum of $k_+^{(\mathcal{D})}$ approximates the one of $k_+$ on $\mathcal{S}$ (see Appendix C.4) so our observations transfer to $k_+$.

**$k_+^{(\mathcal{D})}$ on $\mathcal{S}$ vs. $k_{\text{b}}$ on $\mathcal{S}'$.** For the base kernels $k_{\text{b}}$ and groups $\mathcal{G}$ considered, the spectrum of $k_+^{(\mathcal{D})}$ on $\mathcal{S} = [-1, 1]$ exactly matches that of $k_{\text{b}}$ on the reduced domain $\mathcal{S}'$. This indicates that $k_+^{(\mathcal{D})}$ faithfully incorporates the extra similarities induced by $\mathcal{G}$-invariance: it retains the eigendecay of $k_{\text{b}}$, but as if it were defined on the quotient space $\mathcal{S}/\mathcal{G}$ of effective volume $\text{vol}(\mathcal{S})/|\mathcal{G}|$.[10]

**$k_+^{(\mathcal{D})}$ on $\mathcal{S}$ vs. $k_{\text{avg}}$ on $\mathcal{S}$.** From Figure 4 (middle and right columns) , it is clear that the spectrum of $k_{\text{avg}}$ decays at least as fast as that of $k_+^{(\mathcal{D})}$. They coincide for the RBF kernel and $k_{\text{avg}}$ decays even faster for the Matérn kernel. In principle, this suggests that $k_{\text{avg}}$ should admit tighter information-gain bounds and thus better regret guarantees. However, our empirical results contradict this prediction, as $k_+^{(\mathcal{D})}$ consistently outperforms $k_{\text{avg}}$. This discrepancy highlights the fact that eigendecay alone does not fully explain BO performance, as pointed out in Sections 5 and 6.

### D.2 SCHATTEN NORM INEQUALITIES

While the empirical spectra in Appendix D.1 already highlight a mismatch between eigendecay and observed BO performance, one may ask whether formal inequalities between the operators induced by $k_{\text{avg}}$ and $k_+$ can be established. We record here for completeness that it is possible to control the Schatten class of $k_+$ in terms of the one of $k_{\text{avg}}$.

Assume: $(\mathcal{S}, \mu)$ is a probability space on which a finite group $\mathcal{G}$ acts measurably, and the base kernel $k_{\text{b}}$ is bounded, symmetric, PSD, and nonnegative. Define

$$k_{\text{avg}}(\boldsymbol{x}, \boldsymbol{x}') := \frac{1}{|\mathcal{G}|^2} \sum_{g,g' \in \mathcal{G}} k_{\text{b}}(g\boldsymbol{x}, g'\boldsymbol{x}'), \qquad k_{\max}(\boldsymbol{x}, \boldsymbol{x}') := \max_{g,g' \in \mathcal{G}} k_{\text{b}}(g\boldsymbol{x}, g'\boldsymbol{x}')$$

and $k_+$ as the kernel corresponding to the positive part of $T_{k_{\max}}$: $T_{k_+} = (T_{k_{\max}})_+$.

**Schatten norm interpolation.** Let $H = L^2(\mu)$ be the separable Hilbert space of squared integrable functions on $(\mathcal{S}, \mu)$, $T : H \to H$ a compact operator, and write $s_i(T)$ for the singular values of $T$, i.e. $s_i(T) = \sqrt{\lambda_i(T^*T)}$, arranged in nonincreasing order and counted with multiplicity. The Schatten-$p$ norm is defined as

$$\|T\|_{S_p} := \left( \sum_i s_i(T)^p \right)^{1/p}, \qquad 1 \leq p < \infty, \qquad \|T\|_{S_\infty} := \sup_i s_i(T).$$

---

[10]For a finite group $\mathcal{G}$ of isometries, one indeed has $\text{vol}(\mathcal{S}/\mathcal{G}) = \text{vol}(\mathcal{S})/|\mathcal{G}|$ (Petersen, 2006).

**Lemma 15** (Monotonicity for pointwise kernels)**.** *If two kernels $k, k'$ are bounded and satisfy $0 \leq k \leq k'$ pointwise, then $\|T_k\|_{S_p} \leq \|T_{k'}\|_{S_p}$ for $p = 2, \infty$. If $k$ and $k'$ are also PSD, then $\|T_k\|_{S_p} \leq \|T_{k'}\|_{S_p}$ for $p = 1$ too.*

*Proof.* For $p = \infty$, the Schatten $p$-norm is the operator norm $\|T\|_{\mathrm{op}} = \sup_{\|f\|_H = 1} \|Tf\|_H$. Pointwise $0 \leq k \leq k'$ implies $\|T_k f\|_H \leq \|T_{k'}|f|\|_H \leq \|T_{k'}\|_{S_\infty} \|f\|_H$, so taking the supremum over $\|f\|_H = 1$ yields $\|T_k\|_{S_\infty} \leq \|T_{k'}\|_{S_\infty}$. If $T = T_k$ is the integral operator associated with a nonnegative kernel $k$, then $\|T_k\|_{S_2} = \|k\|_{L^2(\mu \otimes \mu)}$. Hence pointwise $0 \leq k \leq k'$ gives $\|T_k\|_{S_2} \leq \|T_{k'}\|_{S_2}$ for $p = 2$ as well. Finally when $k$ is PSD, we have $\|T_k\|_{S_2} = \int_x k(x, x) d\mu(x)$ (and similarly for $k'$) and again a pointwise comparison yields the result. $\qquad\square$

From this we immediately obtain, for our specific kernels that for $p = 2, \infty$, and also $p = 1$ if $k_{\max}$ is PSD:

$$k_{\mathrm{avg}} \leq k_{\max} \leq |\mathcal{G}|^2 k_{\mathrm{avg}} \quad \Rightarrow \quad \|T_{k_{\mathrm{avg}}}\|_{S_p} \leq \|T_{k_{\max}}\|_{S_p} \leq |\mathcal{G}|^2 \|T_{k_{\mathrm{avg}}}\|_{S_p}$$

**Lemma 16** (Interpolation inequalities for Schatten norms)**.** *For any nonnegative sequence $a = (a_i)_{i \geq 1}$ one has*

$$\|a\|_{\ell^p} \leq \|a\|_{\ell^2}^{2/p} \|a\|_{\ell^\infty}^{1-2/p} \qquad (p \geq 2),$$

$$\|a\|_{\ell^p}^p \leq \|a\|_{\ell^1}^{2-p} \|a\|_{\ell^2}^{2(p-1)} \qquad (1 \leq p \leq 2).$$

*Proof.* For $p \geq 2$, $\sum_i a_i^p = \sum_i a_i^{p-2} a_i^2 \leq \|a\|_{\ell^\infty}^{p-2} \sum_i a_i^2$, giving the stated inequality. For $1 \leq p \leq 2$, write

$$\sum_i a_i^p = \sum_i a_i^{2-p} a_i^{2(p-1)}.$$

Let $r = \frac{1}{2-p}$ and $s = \frac{1}{p-1}$ (with the usual convention $1/0 = \infty$). For $1 < p < 2$ we have $1 < r, s < \infty$ and by Hölder,

$$\sum_i a_i^p \leq \left(\sum_i (a_i^{2-p})^r\right)^{1/r} \left(\sum_i (a_i^{2(p-1)})^s\right)^{1/s} = \left(\sum_i a_i\right)^{1/r} \left(\sum_i a_i^2\right)^{1/s}.$$

Since $1/r = 2 - p$ and $1/s = p - 1$, this gives

$$\|a\|_{\ell^p}^p \leq \|a\|_{\ell^1}^{2-p} \|a\|_{\ell^2}^{2(p-1)}.$$

The endpoint cases $p = 1, 2$ follow by continuity (and are trivial directly). $\qquad\square$

Applied to $a_i = s_i(T)$, Lemma 16 yields the standard Schatten interpolation inequalities:

$$\|T\|_{S_p} \leq \|T\|_{S_2}^{2/p} \|T\|_{S_\infty}^{1-2/p}, \quad (p \geq 2),$$

$$\|T\|_{S_p} \leq \left(\|T\|_{S_1}\right)^{\frac{2}{p}-1} \left(\|T\|_{S_2}^2\right)^{1-\frac{1}{p}}, \quad (1 \leq p \leq 2).$$

Since the spectrum of $T_{k_+}$ is the positive part of the one of $T_{k_{\max}}$, we have $\|T_{k_+}\|_{S_p} \leq \|T_{k_{\max}}\|_{S_p}$. We deduce the next lemma.

**Lemma 17.** *For $p \geq 2$:*

$$\|T_{k_+}\|_{S_p} \leq \|T_{k_{\max}}\|_{S_p} \leq |\mathcal{G}| \|T_{k_{\mathrm{avg}}}\|_{S_2}^{2/p} \|T_{k_{\mathrm{avg}}}\|_{S_\infty}^{1-2/p}$$

*and if $k_{\max}$ is already PSD then for $1 \leq p \leq 2$:*

$$\|T_{k_+}\|_{S_p} = \|T_{k_{\max}}\|_{S_p} \leq |\mathcal{G}| \left(\|T_{k_{\mathrm{avg}}}\|_{S_1}\right)^{2/p-1} \left(\|T_{k_{\mathrm{avg}}}\|_{S_2}^2\right)^{1-1/p}$$

*and*

$$\|T_{k_{\mathrm{avg}}}\|_{S_p} \leq \left(\|T_{k_{\max}}\|_{S_1}\right)^{2/p-1} \left(\|T_{k_{\max}}\|_{S_2}^2\right)^{1-1/p}.$$

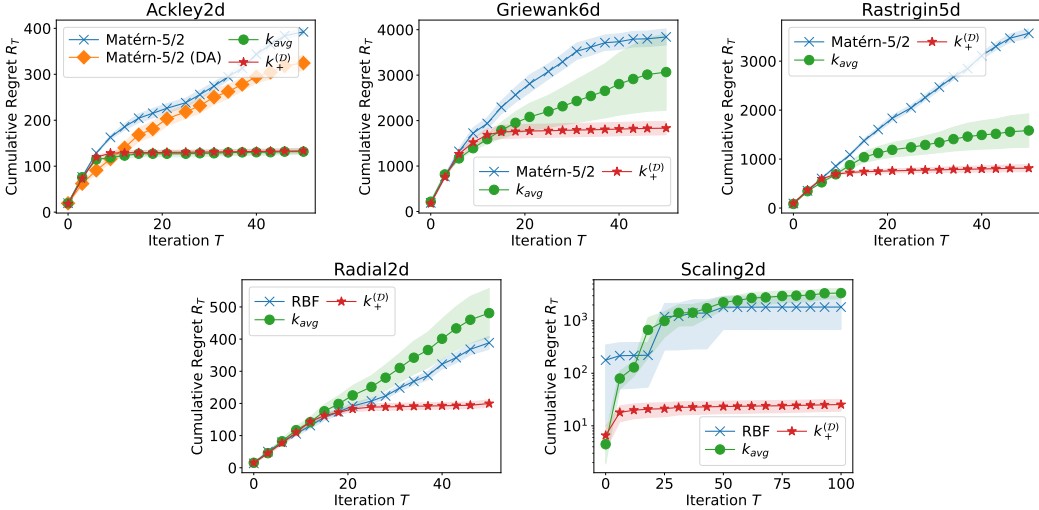

Figure 5: Cumulative regret under GP-UCB with $k_b$ (blue crosses), $k_{avg}$ (orange diamonds), and $k_+^{(\mathcal{D})}$ (green circles) on synthetic benchmarks. Shaded areas: standard error over 10 seeds.

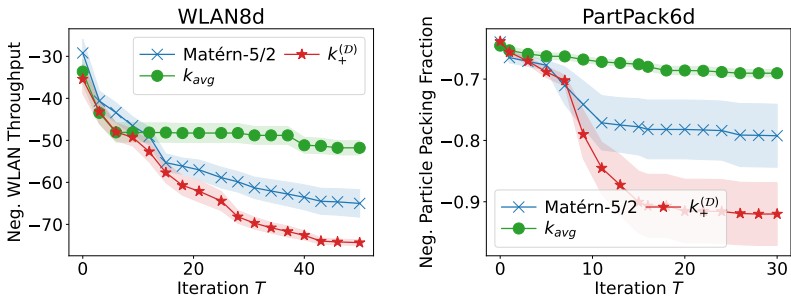

Figure 6: Negated simple reward under GP-UCB with $k_b$ (blue crosses), $k_{avg}$ (orange diamonds), and $k_+^{(\mathcal{D})}$ (green circles) on real-world experiments. Shaded areas: standard error over 10 seeds.

## E  BENCHMARKS

In this appendix, we present additional results and describe the experimental setup of Section 4 in detail.

### E.1  EXPERIMENTAL FIGURES

We provide the whole set of figures generated from our experiments on synthetic benchmarks (Figure 5) and on real-world problems (Figure 6).

### E.2  EXPERIMENTAL DETAILS

In our experiments, every BO algorithm is implemented with the same BO library, namely BOTorch (Balandat et al., 2020). All of them are initialized with five observations sampled uniformly in $\mathcal{S}$. After that, at each iteration $t$, every BO algorithm must:

- **Fit its kernel hyperparameters.** This is done by gradient ascent of the Gaussian likelihood, as recommended by BOTorch. The hyperparameters are the signal variance $\lambda$, the lengthscale $l$ and the observational noise level $\sigma_0^2$.
- **Optimize GP-UCB to find $\boldsymbol{x}_t$.** This is done by multi-start gradient ascent, using the `optimize_acqf` function from BOTorch. As values of $\beta_t$ recommended by Srinivas et al. (2012) turn out to be too exploratory in practice, we set $\beta_t = 0.5d \log(t)$.
- **Observe** $y(\boldsymbol{x}_t) = f(\boldsymbol{x}_t) + \epsilon_t$. Function values are corrupted by noise whose variance is 2% of the signal variance.

We optimize over 50 iterations and typically measure the cumulated regret along the optimizer's trajectory.

All experiments are replicated across ten independent seeds and are run on a laptop equipped with an Intel Core i9-9980HK @ 2.40 GHz with 8 cores (16 threads). No graphics card was used to speed up GP inference. The typical time for each maximization problem ranged from $\sim 1$ minute (two-dimensional Ackley, $|\mathcal{G}| = 8$) to $\sim 15$ minutes (five-dimensional Rastrigin, $|\mathcal{G}| = 3840$). The particle packing problem was by far the most time-consuming experiment due to the expensive physics simulator used for computing the objective value of each new query ($\sim 4$ hours for 30 BO iterations, which we repeated on 10 seeds for each kernel).

### E.3 BENCHMARKS

We maximize the following functions.

**Ackley.** The $d$-dimensional Ackley function $f_{\text{Ackley}}$ on $\mathcal{S} = [-16, 16]^d$ with global maximum $f_{\text{Ackley}}(\mathbf{0}) = 0$, with $-f_{\text{Ackley}}$ defined by:

$$-f_{\text{Ackley}}(\boldsymbol{x}) = -a \exp\left(-b\sqrt{\frac{1}{d}\sum_{i=1}^{d} x_i^2}\right) - \exp\left(\frac{1}{d}\sum_{i=1}^{d} \cos(cx_i)\right) + a + \exp(1), \quad (15)$$

where we set $a = 20$, $b = 0.2$ and $c = 2\pi$ as recommended.

The $d$-dimensional Ackley is invariant to the hyperoctahedral group in $d$ dimensions, which includes permutations composed with coordinate-wise sign-flips. Consequently, in $d$ dimensions, $|\mathcal{G}| = \underbrace{2^d}_{\text{sign-flips}} \underbrace{d!}_{\text{permutations}}$.

**Griewank.** The $d$-dimensional Griewank function $f_{\text{Griewank}}$ on $\mathcal{S} = [-600, 600]^d$ with global maximum $f_{\text{Griewank}}(\mathbf{0}) = 0$, with $-f_{\text{Griewank}}$ defined by:

$$-f_{\text{Griewank}}(\boldsymbol{x}) = \sum_{i=1}^{d} \frac{x_i^2}{4000} - \prod_{i=1}^{d} \cos\left(\frac{x_i}{\sqrt{i}}\right) + 1.$$

The $d$-dimensional Griewank is invariant to coordinate-wise sign-flips of all $d$ coordinates. Therefore, in $d$ dimensions, $|\mathcal{G}| = 2^d$.

**Rastrigin.** The $d$-dimensional Rastrigin $f_{\text{Rastrigin}}$ on $\mathcal{S} = [-5.12, 5.12]^d$ with global maximum $f_{\text{Rastrigin}}(\mathbf{0}) = 0$, with $-f_{\text{Rastrigin}}$ defined by:

$$-f_{\text{Rastrigin}}(\boldsymbol{x}) = 10d + \sum_{i=1}^{d} \left(x_i^2 - 10\cos\left(2\pi x_i\right)\right).$$

The $d$-dimensional Rastrigin is invariant to the hyperoctahedral group in $d$ dimensions, which includes permutations composed with coordinate-wise sign-flips. Consequently, in $d$ dimensions, $|\mathcal{G}| = \underbrace{2^d}_{\text{sign-flips}} \underbrace{d!}_{\text{permutations}}$.

**Radial.** Our radial benchmark is defined on $\mathcal{S} = [-10, 10]^2$ with global maxima $f_{\text{Radial}}(\boldsymbol{x}^*) = 0$, where $\boldsymbol{x}^*$ is any $\boldsymbol{x} \in \mathcal{S}$ such that $||\boldsymbol{x}||_2 = ab$. It has the following expression:

$$f_{\text{Radial}}(\boldsymbol{x}) = f_{\text{Rastrigin}}\left(\frac{||\boldsymbol{x}||_2}{a} - b\right) \quad (16)$$

where we set $a = 10\sqrt{2}$, $b = 0.8$ and where $f_{\text{Rastrigin}}$ is the one-dimensional Rastrigin benchmark.

Our radial benchmark is invariant to planar rotations. Consequently, $\mathcal{G}$ comprises an uncountably infinite number of symmetries.

**Scaling.**  Our scaling benchmark is defined on $\mathcal{S} = [0.1, 10]^2$ with global maxima $f_{\text{Scaling}}(\boldsymbol{x}^*) = 0$, where $\boldsymbol{x}^*$ is any $\boldsymbol{x} = (x_1, x_2) \in \mathcal{S}$ such that $x_1 = x_2$. The function $-f_{\text{Scaling}}$ has the following expression:

$$-f_{\text{Scaling}}(\boldsymbol{x}) = \left( \frac{x_1}{x_2} - 1 \right)^2.$$

Our scaling benchmark is invariant to rescaling of both coordinates. Consequently, $\mathcal{G}$ comprises an uncountably infinite number of symmetries.

**WLAN.**  The goal of the WLAN benchmark is to place $m$ access points (APs) inside a square region $\mathcal{A} = [-50, 50]^2$ so as to maximize the total communication quality over $p$ users located in $\mathcal{A}$, a recurring problem in wireless network design (Younis & Akkaya, 2008; Taleb et al., 2022). Given a set of AP positions, each user connects to its closest AP, and the resulting network throughput—computed from the Signal to Interference plus Noise Ratio (SINR) and Shannon capacities—defines the value of the objective function.

The user positions $\{(u_j, v_j)\}_{j \in [p]} \subset \mathcal{A}$ and all physical parameters ($W$, $L$, $\lambda$, $N$) are given. The region $\mathcal{A}$ itself is fixed.

The variables of the problem are the AP locations

$$(\boldsymbol{x}, \boldsymbol{y}) = ((x_1, \ldots, x_m), (y_1, \ldots, y_m)) \in \mathcal{S} = \mathcal{A}^m,$$

so the search space is $2m$-dimensional. Every quantity below—AP–user associations, distances, received powers, SINRs, and capacities—depends on $(\boldsymbol{x}, \boldsymbol{y})$.

For a candidate placement $\{(x_i, y_i)\}$, each user attaches to its nearest AP. Thus AP $i$ serves the users in

$$\mathcal{U}(x_i, y_i) = \{\, j \in [p] : d_{ij} \leq d_{kj} \text{ for all } k \neq i \,\},$$

(ties are resolved arbitrarily) where the distance to user $j$ is

$$d_{ij} = \sqrt{(x_i - u_j)^2 + (y_i - v_j)^2}.$$

For any associated pair $(i, j)$, the power received by user $j$ from AP $i$ is

$$P_{ij} = 10^{-L/10} \, \min(d_{ij}^{-\lambda}, 1),$$

and the SINR is

$$\gamma_{ij} = \frac{P_{ij}}{N + \sum_{k \neq i} P_{kj}}.$$

The corresponding Shannon capacity is

$$C_{ij} = W \log_2(1 + \gamma_{ij}).$$

Maximizing the WLAN performance amounts to maximizing the total throughput (the cumulated sum of Shannon capacities for every AP-user association):

$$f_{\text{WLAN}}(\boldsymbol{x}, \boldsymbol{y}) = \sum_{i=1}^{m} \sum_{j \in \mathcal{U}(x_i, y_i)} C_{ij}$$

viewed as a function of the AP locations $(\boldsymbol{x}, \boldsymbol{y})$.

In our experiment, we set $W = 1$ MHz, $L = 46.67$ dBm, $\lambda = 3$, $N = -85$ dBm, $m = 4$ APs and $p = 16$ users.

Our objective $f_{\text{WLAN}}$ is invariant to any permutation of the APs: permuting both $\boldsymbol{x}$ and $\boldsymbol{y}$ with the same permutation leaves the objective value unchanged. Therefore, $|\mathcal{G}| = m!$.

Figure 7 shows the best AP-placement found by GP-UCB using $k_+^{(\mathcal{D})}$ on one training run.

**Particle packing problem.** The particle packing fraction (PPF) problem models how a mixture of spherical particles settles under gravity inside a fixed rectangular box. This setting originates from

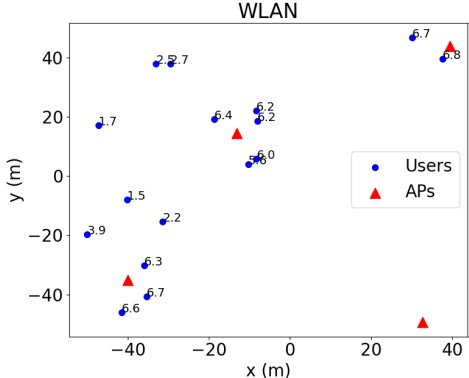

Figure 7: WN with the best positions of APs found by GP-UCB with $k_+^{(\mathcal{D})}$. APs are depicted by red triangles and users with blue circles. The throughput for each user is shown in Mbps.

granular-material physics and is routinely used in materials science and civil engineering (e.g., in the design of concrete mixes (Li et al., 2023; Basheerudeen & Anandan, 2014) by tuning the size distribution and proportions of aggregates to maximize packing density; for instance to need less cement and water, and get better mechanical properties).

People literally design concrete mixes by tuning the size distribution and proportions of aggregates to maximize packing density (so you need less cement and water, and get better mechanical properties).

In this problem, a mixture of particles is first instantiated inside the box according to prescribed mixture parameters, and the particles are then allowed to fall under gravity. Collisions, frictions and rearrangements determine the final configuration, and the packing fraction is defined as the ratio between the total particle volume and the volume of the smallest axis-aligned box that contains all particles after settling.

We fix the number of particle types to $n$. Each type $i$ is described by:

- a diameter $d_i$ in a prescribed interval $[d_{\min}, d_{\max}]$,
- a share $s_i$ in $[s_{\min}, s_{\max}]$, representing the relative proportion of particles of that type in the mixture.

Thus the optimization variable is

$$\boldsymbol{x} = (d_1, \ldots, d_n, \ s_1, \ldots, s_n).$$

The box size and the total initial particle volume $V_p$ (which then remains constant during the simulation) are fixed in all experiments.

Given a mixture specification $\boldsymbol{x} = (d_1, \ldots, d_n, \ s_1, \ldots, s_n)$, the initial particle configuration is generated by repeatedly sampling particles until a fixed total particle volume $V_p$ is reached. Particles are sampled independently as follows: (i) sample a type $i \in \{1, \ldots, n\}$ with probability proportional to its share $s_i$, (ii) sample a location uniformly at random in the container and put a particle of diameter $d_i$ there. If any overlap of particles occurs during initialization, positions are adjusted locally so that the configuration becomes valid. From this randomized initial state, the system evolves under gravity, in practice we use a physics-based simulator (LAMMPS (Thompson et al., 2022)) for that. The simulation proceeds until the particles reach a mechanically stable configuration, as illustrated in Figure 8. If $V_o(\boldsymbol{x})$ denotes the volume of the smallest axis-aligned box enclosing all particles at the end of the dynamics (i.e., the container volume after settling), the particle packing fraction is

$$\text{PPF}(\boldsymbol{x}) = \frac{V_p}{V_o(\boldsymbol{x})},$$

and we aim at maximizing this as a function of the mixture parameters $\boldsymbol{x}$. To our knowledge, there is no accurate closed-form expression for this dynamical packing fraction in our setup, so evaluating $\text{PPF}(\boldsymbol{x})$ requires running the full physical simulation. Indeed: $\text{PFF}(\boldsymbol{x})$ is actually a random variable: given any mixture parameters $\boldsymbol{x}$, $V_o(\boldsymbol{x})$ depends on the random initialization of the particles in the container, so there is observational noise induced by this random initialization. Moreover, even if the random seed was fixed, because $V_o(\boldsymbol{x})$ depends on complex interactions during the fall—collisions, friction, and rearrangements, there is still no closed form available: evaluating $\text{PPF}(\boldsymbol{x})$ always

requires running this full physical simulation. This makes the objective function costly and genuinely black-box, a typical regime where BO is well motivated.

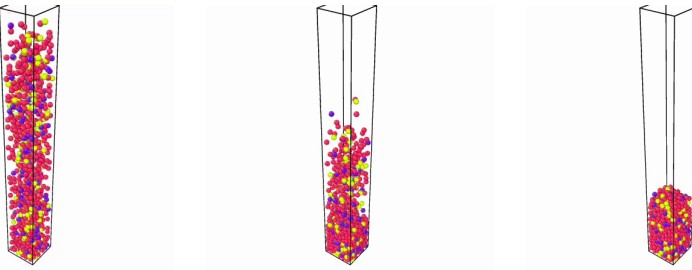

Figure 8: Particles settling under gravity in a fixed-size box. A single evaluation of $\mathrm{PPF}(\boldsymbol{x})$ requires simulating the fall from a randomized initial configuration (left) to a mechanically stable state (right), making the objective expensive and simulation-based.

Two symmetries are inherent to this formulation:

1. *Share scaling:* multiplying all $s_i$ by the same positive factor leaves the resulting mixture unchanged (the mixture only involves normalized shares).
2. *Permutation symmetry:* permuting the $(d_i, s_i)$ pairs does not change the mixture either.

In practice, we take $n = 3$, which is the smallest setting where the problem starts to be interesting (no easy solution) while keeping simulation costs manageable. We constrain the diameters and shares to

$$d_i \in [0.35,\ 0.80], \qquad s_i \in [0.1,\ 1.0],$$

chosen so that (i) all particles remain sufficiently small relative to the fixed box size, and (ii) each type is represented in non-negligible quantity.

Baird et al. (2023a) previously applied BO to this problem (for solid rocket fuel design) and handled these symmetries by restricting the search to a fundamental domain and applying standard kernels there. In contrast, we keep the domain unchanged and instead use kernels that are *invariant* under the symmetries of the problem. A conceptual comparison between these two symmetry-handling strategies is provided in Appendix G.

## F    COMPARISON OF SYMMETRY-INVARIANT KERNELS WITH THE DATA-AUGMENTATION APPROACH

Given the widespread use of data augmentation (DA), we compare symmetry-invariant kernels with the simple baseline corresponding to using the base kernel combined with DA. We find that symmetry-invariant kernels perform better overall.

DA consists of replacing each input $x$ in the dataset by $(gx)_{g \in \mathcal{G}'_x}$ with $\mathcal{G}'_x \subset \mathcal{G}$, and BO is run on this augmented dataset. We consider two scenarios: (i) using all augmentations for small groups ($\mathcal{G}'_x = \mathcal{G}$ for all $x$) so that $(gx)_{g \in \mathcal{G}'}$ is simply the orbit of $x$, and (ii) using a random subset $\mathcal{G}'_x \subset \mathcal{G}$ for larger groups (chosen independently for every $x$, drawn uniformly without replacement).

On the two-dimensional Ackley function (left panel of Figure 9), $k_b$ is applied to a dataset augmented with all symmetries ($|\mathcal{G}| = 8$). In this case, $k_b$ with DA achieves slightly better (lower) cumulative regret than $k_b$ alone. Its performance, however, remains worse that of the average kernel $k_{\mathrm{avg}}$ and the PSD projection of the max kernel $k_+^{(\mathcal{D})}$. A similar pattern appears on the three-dimensional Ackley function (right panel of Figure 9), where DA uses 20 augmentations sampled without replacement from $\mathcal{G}$ ($|\mathcal{G}| = 48$).

We also report the runtime of each method. These results show that $k_b$+DA scales less favorably than $k_{\mathrm{avg}}$ and $k_+^{(\mathcal{D})}$, even when using only a moderate random subset of augmentations. Overall, these experiments suggest that using symmetry-invariant kernels directly is more practical for Bayesian optimization than relying on data augmentation.

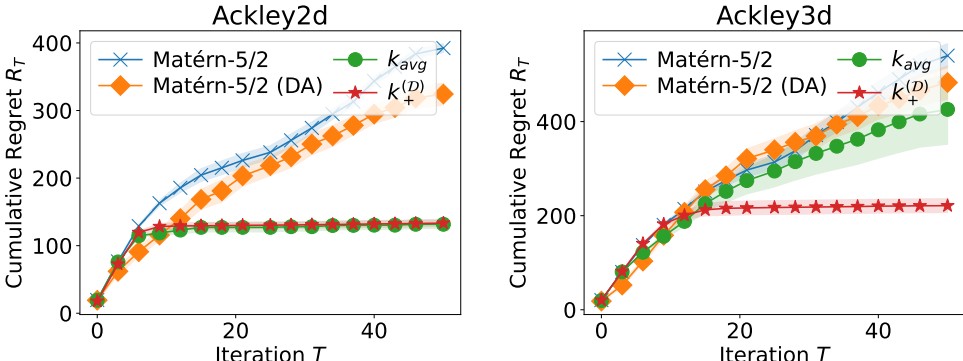

Figure 9: Cumulative regret on the two-dimensional (resp., three-dimensional) Ackley function, with $|G| = 8$ (resp., $|G| = 48$).

Table 3: Average wall-clock time in seconds per iteration for each method on the two-dimensional (resp., three-dimensional) Ackley function.

| Benchmark | $|G|$ | $k_b$ | $k_b$ with DA | $k_{\mathrm{avg}}$ | $k_+^{(\mathcal{D})}$ |
|---|---|---|---|---|---|
| Ackley2d | 8 | $0.416 \pm 0.253$ | $0.599 \pm 0.279$ | $0.451 \pm 0.273$ | $0.924 \pm 0.444$ |
| Ackley3d | 48 | $0.506 \pm 0.336$ | $2.665 \pm 2.950$ | $0.590 \pm 0.384$ | $1.307 \pm 0.724$ |

# G  WORKING WITH FUNDAMENTAL DOMAINS AND QUOTIENTS

This appendix expands on the brief discussion in Section 2.2 about search-space restriction and explains why our approach targets kernel design rather than the choice of domain. The goal is to clarify that both ingredients, a good domain and a good kernel, are needed and complementary.

## G.1  FUNDAMENTAL DOMAINS AS QUOTIENT REPRESENTATIONS

Given a domain $\mathcal{S}$ and a group action $\mathcal{G}$, restricting the search to a fundamental domain amounts to choosing a concrete embedded representation of the quotient space $\mathcal{S}/\mathcal{G}$ in $\mathcal{S}$. While this is conceptually elegant, the practical implementation depends heavily on the pair $(\mathcal{S}, \mathcal{G})$ and must be re-derived for each new problem.

## G.2  EXAMPLE: PERMUTATIONS OF $\mathbb{R}^d$

In several of our experiments, $\mathcal{S} = [a, b]^d$ and $\mathcal{G} = S_d$ acts by permuting coordinates. Two vectors are equivalent if one is a permutation of the other. A natural choice of fundamental domain is the *sorted cone*

$$\mathcal{C} = \{x \in [a, b]^d : x_1 \le x_2 \le \cdots \le x_d\},$$

which is one possible representation of the quotient $\mathcal{S}/\mathcal{G}$ (other equivalent views include multisets or $d$-atomic probability measures, but these views does not lead to subsets of the original domain $\mathcal{S}$ so they do not qualify as "fundamental domains").

Even in this simple case, two practical issues appear.

*(1) One must characterize and project onto the quotient, and check that it is "smooth enough".* Most BO implementations assume that the search domain is a box $[a, b]^d$ for which enforcing feasibility of the iterates is straightforward (via coordinatewise clipping $x \mapsto \max(a, \min(b, x))$). If we optimize an acquisition function over the fundamental domain $\mathcal{C}$ instead, any gradient-based or heuristic optimizer will typically propose points $x$ that lie outside $\mathcal{C}$, and these must be projected back. This requires (i) describing the quotient $\mathcal{S}/\mathcal{G}$ via an explicit embedded representation (here, $\mathcal{C} \subset \mathcal{S}$) and (ii) figuring out how to implement the projection. For $\mathcal{C}$, projecting $x$ onto it amounts to solving

$$\mathrm{proj}_{\mathcal{C}}(x) \ \in \ \arg \min_{y_1 \le \cdots \le y_d} \|y - x\|^2,$$

which can be solved efficiently using known algorithms (e.g. the pool adjacent violators algorithm). Our point is not that this particular projection is hard, but that for each new pair $(\mathcal{S}, \mathcal{G})$ the user must

again derive an explicit model of the quotient and a practical projection operator, which can be a burden depending on their goals and familiarity with quotients and the problem at hand.

*Smoothness assumptions also need to be checked.* The cone $\mathcal{C}$ is not a smooth manifold, implying that the projection is not smooth everywhere and gradients are not smooth (or even properly defined) at certain points. Here, the singularities form a zero-measure set: they occur at points with some equal coordinates (this is because the action of $\mathcal{G}$ is not free; in contrast, if the action were free, proper, and smooth, Theorem 21.10 in Lee (2013) would guarantee that the quotient is a smooth manifold). For many constrained sets, singularities similarly form a negligible set and may be harmless for optimization (initialization and gradient descent are likely to avoid them), but this depends on the specific quotient and must be verified on a case-by-case basis.

Overall, working in the quotient means that the user must (i) characterize and project onto a potentially non-smooth quotient, and (ii) check that its singularities do not cause difficulties for the optimization method they use. Doing this for each new $(\mathcal{S}, \mathcal{G})$ may be burdensome. This is why, in this paper, we choose to avoid optimizing in a fundamental domain and instead provide kernels that can be used in a plug-and-play manner directly on $\mathcal{S}$. These same kernels could also be used on the quotient space (by interpreting them as kernels on equivalence classes), so our approach is complementary to, rather than in competition with, the choice of the search domain.

*(2) One must still choose a kernel on equivalence classes.* Working on $\mathcal{S}/\mathcal{G}$ does not remove the modelling choice: one still needs to pick a kernel $k([x], [y])$, and there is no canonical option even in the permutation example. The quotient can be described in several equivalent ways (sorted vectors in $\mathcal{C}$, multisets, or atomic measures), and each viewpoint naturally suggests different classes of kernels or distances. This is precisely the type of question our paper addresses: how to construct a good kernel that is invariant to the symmetries? We study a natural construction: start with a "good" kernel on $\mathcal{S}$ (e.g. one that makes sense locally on $\mathcal{S}$ to measure similarity before accounting for symmetries), and then make it invariant by aggregating via mean or max. The resulting kernels are $\mathcal{G}$-invariant and thus well-defined on the quotient, and our results show that the max-based construction shows good properties, both empirically and geometrically.

## H    USE OF LLMS

We made limited use of large language models (GPT-5) during the preparation of this manuscript. Their role was strictly restricted to grammar correction, improving clarity and conciseness, emphasizing text (e.g., bolding), and formatting tables. They were not used for generating technical content, suggesting new concepts, or contributing to proofs or results. All ideas, proofs, experiments, and findings are entirely our own. Every rephrased passage was carefully reviewed and validated by the authors to ensure correctness and faithfulness to our original intent. No unverified or plagiarized content was introduced.

