# OpenReview forum: "Symmetry-Aware Bayesian Optimization via Max Kernels"
_ICLR.cc/2026/Conference — ICLR 2026 Poster_

### Official Review · Reviewer_mZAc · 2025-10-30

**Soundness:** 4
**Presentation:** 4
**Contribution:** 3
**Rating:** 8
**Confidence:** 3

**Summary:**

The authors claim to contribute:
- Introduce a PSD projection of the max-alignment kernel over a symmetry group to avoid typical kernel averaging strategies, which can dilute high-similarity alignments when the symmetry group is large. This kernel is a valid GP kernel and doesn’t change the asymptotic cost of GP inference, so can be used for BO when the objective function is invariant under a known symmetry group.
- Their max kernel behaves as one expects on toy problems and does really well on standard BO test cases in comparison to the average kernel, which is the closest invariant kernel in the literature so far. It also outperforms standard kernels.

**Strengths:**

- Paper is overall well written, clear, and concise. It outlines relevant related work and explains why their method might be preferred to each, situating their method well within the literature. Exposition of method is clear.
- Method seems general—can be used with any base kernel and be PSD thanks to their construction. Can be used with any symmetry group, just need to maximize over orbits. This operation scales quadratically with the size of the symmetry group but this is a problem for previous invariant kernels too (e.g., average kernels).
- Clear 1-D examples showcase how the max kernel more accurately models the objective than the average kernel.
- Regret is impressive compared to base and average kernels.
- Compares findings with spectral analyses and finds a gap, and suggests new ways to explain this gap, which may lead nicely into future work.

**Weaknesses:**

- I would appreciate more real-world BO test cases that are group invariant beside the one WLAN problem, to demonstrate that this is actually an important innovation for applications people care about. More argument about the relevance of this area in the intro, e.g., examples, could also help.


General feedback for improving paper: I think the intro is a little abrupt and a smoother motivation of why we care about max alignment before introducing notation and the PSD projection of the max kernel would help the reader.


I strongly recommend acceptance — the paper is well-written, introduces a max-alignment kernel which is not in the BO literature yet (to best of my knowledge) by approximating it with a projection to make it PSD. This leads to really good results on both toy, interpretable settings and on standard (group invariant) BO test functions. It seems clear that this work would be interesting to the BO community.

**Questions:**

I’m not an expert; why is it difficult to identify a suitable fundamental domain to restrict the search space to, if one knows the symmetry group? Is there an easy example to demonstrate this? It would be good to explain this to readers less familiar with symmetry (which includes much of the BO community).

---

> ### Author Response · Authors · 2025-11-19
>
> We thank the reviewer for the valuable feedback. We answer the raised points below (and we acknowledge the reviewer in the acknowledgement section of the paper). Happy to discuss more if needed!
>
> ## More real-world BO problems and smoother motivation in the intro
>
> Thank you for the suggestions. We used the extra page to clarify the motivation behind the max-alignment idea in the introduction. We also hope to add another real-world example in the revision: we are currently running experiments on a particle-packing problem using a state-of-the-art simulator (but this simulator is quite expensive to run, so results shall take a few days). We hope to come back soon with results.
>
> ## Clarify why identifying a fundamental domain is difficult
>
> We thank the reviewer for the helpful suggestion. We have added an explicit example in Appendix G (with a pointer from section 2.2) illustrating the challenges of identifying and working with a fundamental domain. This also helped us clarify how we position the contribution of the paper: our approach does not compete with methods that optimize directly over a fundamental domain. Instead, it complements these methods as it helps choose a kernel in the fundamental domain.
>
> For BO, one must choose (i) a search domain $S$ and (ii) a kernel on that domain. Given a group of symmetries $G$, even if one decides to optimize over a fundamental domain (i.e., the quotient space $S/G$), a kernel is still required. Kernels on $S/G$ correspond precisely to invariant kernels on the original space $S$. Our work focuses precisely on how to construct a good invariant kernel. Now, we carry out experiments deliberately in $S$ for the reasons detailed in Appendix G: (i) benchmarking: standard BO benchmarks operate on the original domain, hence we do the same to enable direct comparison, and (ii) plug-and-play deployment: our kernels are meant to be plug-and-play on any new pair $(S,G)$, without requiring the user to identify $S/G$, assess its smoothness, and implement a projection map onto it for each new $(S,G)$, which can be cumbersome in practice.

---

> > ### Author Response · Authors · 2025-11-27
> >
> > A quick update: the particle-packing experiment we mentioned earlier is now complete, and the results are included in Section 4.2 and Appendix E.3. They align with our previous findings: max-alignment performs best.

---

### Official Review · Reviewer_4Tfz · 2025-11-01

**Soundness:** 4
**Presentation:** 4
**Contribution:** 3
**Rating:** 8
**Confidence:** 3

**Summary:**

The paper introduces a Gaussian process kernel that handles symmetries (invariance over a group action), for the purpose of Bayesian optimization. One approach is to average the kernel over the symmetries. This paper proposes instead taking a max, which is not necessarily PSD, so then a straightforward PSD projection is included as well.

The paper does a theoretical analysis that provides strong motivation for the use of the max rather than the average over the group for the kernel. Empirical results show that this new kernel performs much better than baseline approaches for handling symmetries. Spectral analysis shows that existing generalization bounds do not predict the improved performance, and the discussion raises ideas for why that might be.

**Strengths:**

* The paper is very clear and is well written.

* The paper addresses an important issue (the detrimental effect of symmetries on GP modeling with standard kernels, and thus on BO performance). The core idea to take the max instead of the average over group orbits is not novel in the general case, but its application to Bayesian optimization is novel, and the paper provides a significant amount of useful analysis to go along with this that make it, in my mind, a very original work.

* I especially appreciated the strong theoretical motivation in Section 3.

* The paper seems particularly appropriate for ICLR as it is about representation learning with symmetries.

**Weaknesses:**

In Section 2.2 the paper describes data augmentation as "a popular way to enforce symmetry." I certainly appreciate that it is intractable in many settings, but by virtue of being popular, it should be compared to in this work. In Ackley2d it could be done directly, since |G| is only 8, and so with 50 total iterations, it becomes a lot of data but still feasible. On other problems, the popular strategy would be to randomly sample a few (still 8 perhaps?) items from G and just augment with those. These are pretty easy to implement and would strengthen the results of the paper significantly. It should come paired with including results showing wall times; the paper suggests that k_avg and k_max will have similar wall time, but it does not actually compare them both to k_b, and so it isn't clear if increasing the amount of data by 8x will make k_b slower than k_max and k_avg.

**Questions:**

N/A

---

> ### Author Response · Authors · 2025-11-19
>
> We thank the reviewer for the valuable feedback. We answer the raised point below (and we acknowledge the reviewer in the acknowledgement section of the paper). Happy to discuss more if needed!
>
> ## Comparison to data augmentation and runtime baselines
>
> We thank the reviewer for this suggestion. In the revision (Appendix F), we now report the performance of the Data Augmentation (DA) baseline using the base kernel $k_b$, evaluated in the two settings proposed by the reviewer: (i) using all augmentations for small groups, and (ii) using a random subset of augmentations for larger groups.
>
> On the two-dimensional Ackley function (left panel of Figure 6), $k_b$ is applied to a dataset augmented with all symmetries ($|G| = 8$). We observe that $k_b$+DA achieves a slightly lower cumulative regret than $k_b$ alone. However, its performance remains substantially worse than that of the mean kernel $k_m$ and the PSD projection of the max kernel $k_+^{(\mathcal{D})}$. The same pattern holds on the three-dimensional Ackley function (right panel of Figure 6), where DA uses 20 augmentations sampled without replacement from $G$ ($|G| = 48$).
>
> Following the reviewer’s suggestion, we also report the runtime of each method in Table 3 of the revision. These results highlight that $k_b$+DA scales also worse than $k_m$ and $k_+^{(\mathcal{D})}$ with the size of the group, even if we scale by using a moderately sized random subset of all possible augmentations. Overall, the experiments confirm the advantage of employing symmetry-invariant kernels directly rather than relying on data augmentation for BO.

---

> > ### Comment · Reviewer_4Tfz · 2025-11-19
> > **Response**
> >
> > Appreciate the new results!

---

### Official Review · Reviewer_6LKt · 2025-11-01

**Soundness:** 3
**Presentation:** 2
**Contribution:** 3
**Rating:** 6
**Confidence:** 2

**Summary:**

This paper proposes Symmetry-Aware Bayesian Optimization (SABO), a framework for exploiting known or learnable symmetries in black-box optimization problems to improve sample efficiency. Traditional approaches, such as Brown et al. (NeurIPS 2024), incorporate known invariances into Gaussian Processes through orbit-averaged kernels. This paper introduces a max-alignment view that aligns input pairs via the most similar symmetry transformations.
To address the fact that the resulting kernel may violate positive semi-definiteness, the authors introduce the projected-max kernel, which applies a PSD projection based on spectral (eigenvalue) analysis to restore validity.
Overall, the paper contributes a more general method for incorporating symmetry structure into BO, extending symmetry exploitation beyond fixed and known invariance groups to settings where symmetries can be learned or adaptively aligned.

**Strengths:**

The paper presents an original and conceptually appealing extension of symmetry-exploiting Bayesian optimization methods, offering a practical and theoretically grounded approach that goes beyond prior invariant-kernel formulations. By introducing the max-alignment view and the projected-max kernel, the authors propose a framework allowing Bayesian optimization to leverage approximate or continuous symmetries without requiring exact group knowledge. The integration of spectral analysis and PSD projection is technically sound and shows thoughtful consideration of kernel stability, while the empirical results convincingly demonstrate the benefits of sharper, symmetry-aware modeling.

**Weaknesses:**

This paper's presentation significantly limits accessibility and clarity, especially for a broader audience. It does not clearly explain why PSD matters and what is the motivation of spectral analysis. Similarly, although the paper includes justification of the max-alignment view, the explanation remains largely heuristic. It does not provide an analytical argument or formal characterization of when or why this leads to better Bayesian optimization performance. As a result, the motivation still feels conceptually underdeveloped and difficult to follow for readers without deep kernel-theoretic background.

**Questions:**

Repeat from the weakness section:
1. Why PSD projection matters?
2. Why consider a spectral analysis?
3. The authors can explain more about how kernel validity relates to PSD properties, and why “valid covariance” matters in GP inference, and what specific advantages “max alignment” brings in the Bayesian optimization setting (e.g., in terms of acquisition behavior, uncertainty modeling, or convergence properties).

---

> ### Author Response · Authors · 2025-11-19
>
> We thank the reviewer for the valuable feedback. We answer the raised points below (and we acknowledge the reviewer in the acknowledgement section of the paper). Happy to discuss more if needed!
>
> ## Motivation for PSD projection and spectral analysis
> We agree that the motivation could be clearer for a general audience. We added what follows in the revision (cf. "Why PSDness of $k$ matters" in section 2.1, and  paragraph "Relation with spectral-based theory" in the introduction).
>
> **Why PSD matters?** There’s no technical impossibility in running BO with a non-PSD kernel, but we then lose all probabilistic interpretations and theoretical guarantees. In particular, the Gaussian Process (GP) posterior mean and variance used to guide exploration/exploitation might behave unpredictably. That’s why we strongly motivate keeping PSDness.
>
> To give a quick picture, a BO loop does the following:
> 1. build the Gram matrix $K(X,X)=(k(x_i,x_j))_{1\le i,j\le n}$,
>
> 2. compute the inverse of $K(X,X) + \sigma^2 I$ (fixed hyperparameter $\sigma$) and use that inverse to define functions $\mu(x), \sigma(x)^2$ (corresponding to the GP posterior mean and variance under the PSD assumption)
>
> 3. choose the next query $x_{n+1}$ by optimizing an acquisition function based on $\mu(x)$ and $\sigma^2(x)$ (the acquisition function is supposed to balance exploration and exploitation in a meaningful way under the PSD assumption).
>
> If $K$ is not PSD, step (2) may fail (the matrix can be non-invertible). Easy fixes are to replace the inverse with a pseudo-inverse or to increase $\sigma$. With the pseudo-inverse, however, the “posterior variance’’ loses its PSD property. Recall that
> $
> \sigma^2(x) = k(x,x) - k(x,X) K(X,X)^{+} ,k(X,x),
> $
> where $K(X,X)^{+}$ is the Moore–Penrose pseudo-inverse. When $K(X,X)$ is indefinite, the bilinear operator
> $
> C(x,x') = k(x,x') - k(x,X) K(X,X)^{+} k(X,x')
> $
> need not define a positive semidefinite covariance function. Consequently, $\sigma^2(x)$ itself can become negative for some $x$. In such regions the uncertainty is ill-defined (one cannot take $\sqrt{\sigma^2(x)}$, which several acquisition functions require), and directions associated with negative values may appear artificially favorable for exploration. This makes the resulting exploration–exploitation behavior difficult to justify and possibly arbitrary. Alternatively, one can inflate $\sigma$, but then the variance becomes nearly flat everywhere and the BO loop degenerates into a blind exploitation. Neither fix preserves the exploration–exploitation balance that makes BO effective in the first place.
> So it’s not impossible, just poorly motivated: all quantities lose their probabilistic meaning and the useful balance between exploration and exploitation disappears.
>
> **Why spectral analysis?**
> Because most mainstream BO guarantees are expressed in terms of the eigendecay of the kernel, a natural question is whether the performance of our kernel can be
> explained through this lens. Interestingly, our findings indicate the opposite, which is why we included this spectral analysis in our paper: we find that the PSD-max kernel typically exhibits *slower* eigendecay than the mean kernel, yet performs *better* in practice. This suggests that eigendecay alone does not capture the advantages of a PSD-max kernel. In the conclusion, we discuss two likely explanations. First, geometric effects: the mean kernel often misaligns its eigenvectors with the directions that matter for optimization, whereas the max-based construction is well-aligned with the geometry of the problem (Figures 1 and 2). We hypothesize that fast eigendecay is useful only when the dominant eigenfunctions are well-aligned with the relevant directions for optimization; otherwise, spectral rates miss essential structure. Second, approximation hardness: the black-box function $f^\star$ may be much harder to approximate in the RKHS of the mean kernel than in the PSD-max one, which can strongly affect BO performance. Together, these observations indicate that the explanation of the gains of the PSD-max kernel goes beyond pure spectral decay.
>
> ## Heuristic justification for max alignment and its benefits for BO
>
> You’re right that the “max alignment” idea is heuristic, and indeed this heuristic does translate into concrete benefits for BO, which we observe throughout the paper. The resulting kernel is geometrically better aligned with the true structure of the problem (Figures 1-2). In practice, this makes
>
> (i) the acquisition function more faithful as it avoids redundant exploration of points that are already explored up to symmetry, and
>
> (ii) uncertainty modeling also more faithful: it gains confidence in unexplored regions that correspond to symmetry-equivalent points, as mentioned in the revised introduction.

---

> > ### Comment · Reviewer_6LKt · 2025-11-24
> > **Additional comments on the figures**
> >
> > Thanks for your detailed explanations and improvement in clarity in the manuscript. I have some additional comments on the figures:
> >
> > 1. I suggest you increasing the font sizes of the x and y labels and also adding some texts to them rather than just leaving alone the math notations.
> >
> > 2. Why Figure 3 and Figure 4 use different error bars?

---

> > > ### Author Response · Authors · 2025-11-27
> > >
> > > Thank you for the suggestions! We have made the suggested changes.

---

### Meta-Review · Area_Chair_eG22 · 2026-01-09

**Summary:**

The reviewers requested some additional baselines and clarification on minor technical details (e.g., why the kernel must be positive definite). These were all generally addressed, and some reviewers explicitly indicated as such.

Some minor comments of my own:
- I think the new text explaining why the kernel being PSD/PD is longer than necessary really -- I would just point out that BO often uses GP surrogates, which require positive definiteness for all sorts of reasons.
- The text in lines 255-263 is italics as a hold over from the example definition.
- Figure 1 is extremely nice for comparing avg to max, and I wish I had seen it earlier. I would honestly move it to the top of section 3. It completely removes any doubt about why you would want to build the max version of the kernel.

**Reviewer Concerns:**

Nothing really outstanding left -- the authors added the new results asked for.

**Reviewer Scores:**

Probably about the same, perhaps 6LKt increases slightly.

---

### Decision · Program_Chairs · 2026-01-26

Accept (Poster)